# A Probabilistic Model behind Self-Supervised Learning

**Alice Bizeul**[*]                                          *alice.bizeul@inf.ethz.ch*
*Department of Computer Science & ETH AI Center, ETH Zurich*

**Bernhard Schölkopf**                                       *bs@tuebingen.mpg.de*
*Max Planck Institute for Intelligent Systems, Tübingen*

**Carl Allen**                                               *carl.allen@ai.ethz.ch*
*Department of Computer Science & ETH AI Center, ETH Zurich*

**Reviewed on OpenReview:** *https: // openreview. net/ forum? id= QEwz7447tR*

## Abstract

In self-supervised learning (SSL), representations are learned via an auxiliary task without annotated labels. A common task is to classify augmentations or different modalities of the data, which share semantic *content* (e.g. an object in an image) but differ in *style* (e.g. the object's location). Many approaches to self-supervised learning have been proposed, e.g. SimCLR, CLIP, and DINO, which have recently gained much attention for their representations achieving downstream performance comparable to supervised learning. However, a theoretical understanding of self-supervised methods eludes. Addressing this, we present a generative latent variable model for self-supervised learning and show that several families of discriminative SSL, including contrastive methods, induce a comparable distribution over representations, providing a unifying theoretical framework for these methods. The proposed model also justifies connections drawn to mutual information and the use of a "projection head". Learning representations by fitting the model generatively (termed *SimVAE*) improves performance over discriminative and other VAE-based methods on simple image benchmarks and significantly narrows the gap between generative and discriminative representation learning in more complex settings. Importantly, as our analysis predicts, SimVAE outperforms self-supervised learning where style information is required, taking an important step toward understanding self-supervised methods and achieving task-agnostic representations.[1]

## 1 Introduction

In self-supervised learning (SSL), a model is trained to perform an auxiliary task without class labels and, in the process, learns representations of the data that are useful in downstream tasks. Of the many approaches to SSL (e.g. see Ericsson et al., 2022), contrastive methods, such as InfoNCE (Oord et al., 2018), SimCLR (Chen et al., 2020a), DINO (Caron et al., 2021) and CLIP (Radford et al., 2021), have gained attention for their representations achieving downstream performance approaching that of supervised learning. These methods exploit semantically related observations, such as different parts (Mikolov et al., 2013; Oord et al., 2018), augmentations (Chen et al., 2020a; Misra & Maaten, 2020), or modalities/views (Baevski et al., 2020; Radford et al., 2021) of the data, considered to share latent semantic *content* (e.g. the object in a scene) and differ in *style* (e.g. the object's position). SSL methods are observed to "pull together" representations of semantically related samples relative to those chosen at random and there has been growing interest in formalising this to explain why SSL methods learn useful representations, but a mathematical mechanism to justify their performance remains unclear.

Figure 1: **SSL Model** for $J$ semantically related samples (terms explained in §3)

---

[*]Correspondence to alice.bizeul@inf.ethz.ch and carl.allen@ai.ethz.ch

[1]The code to reproduce SimVAE can be found at https://github.com/alicebizeul/simvae

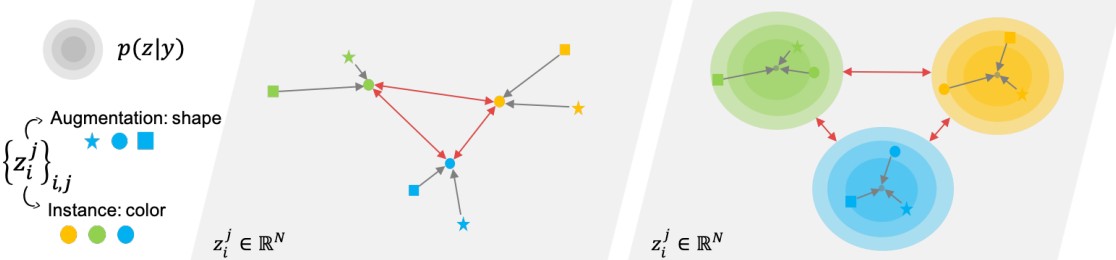

Figure 2: **The representation space $\mathcal{Z}$ of self-supervised representation learning.** (*left*) common view: representations of pairs of semantically related data (e.g. augmentations of an image) are "pulled together" (grey arrows), while other pairs "pushed apart" (red arrows); (*right*) probabilistic view: representations of semantically related data are samples from a common mixture component, $z_i^j \sim p(z|y=i)$. Maximising $\text{ELBO}_{\text{SSL}}$ pulls $z_i^j$ towards a mode for each $i$ (grey dots), clusters are held apart by the need to reconstruct.

We propose a principled rationale for multiple self-supervised approaches, spanning instance discrimination (Dosovitskiy et al., 2014), deep clustering (Caron et al., 2018) and contrastive learning, including the popular InfoNCE loss (Chen et al., 2020a; Radford et al., 2021) (referred to as *predictive* SSL). We draw a connection between these theoretically opaque methods and fitting a latent variable model by variational inference. More specifically, we: i) treat representations as latent variables and vice versa; ii) propose the SSL Model (Fig. 1) as a generative latent variable model for SSL and derive its evidence lower bound ($\text{ELBO}_{\text{SSL}}$); and iii) show that the discriminative loss functions of predictive SSL algorithms approximate the generative $\text{ELBO}_{\text{SSL}}$, up to its generative reconstruction term.

Under the SSL Model, data is assumed to be generated by sampling: i) a high-level latent variable $y \sim p(y)$, which determines the semantic *content* of the data; ii) a latent variable $z \sim p(z|y)$, which governs the *style* of the data and gives rise to differences between semantically related samples (i.e. with same $y$); and iii) the data $x \sim p(x|z)$. Under simple Gaussian assumptions, latent variables of semantically related data form clusters, $z^j \sim p(z|y)$, $j \in [1, J]$ (Fig. 2, *right*), mirroring in a principled way how SSL objectives "pull together" representations of semantically related data and "push apart" others (Wang & Isola, 2020) (Fig. 2, *left*).

We derive the evidence lower bound, $\text{ELBO}_{\text{SSL}}$, as a training objective to fit the SSL Model and show that it closely relates to the loss functions of discriminative predictive SSL methods. Thus, predictive SSL methods induce comparable latent structure to the mixture prior $p(z)$ of the SSL Model, but differ in that they encourage latent clusters $p(z|y)$ to "collapse" and so lose style information that distinguishes semantically related data samples (see Figures 3 and 4). Given that a downstream task may require style information, e.g. for object localisation, this could limit the generality of SSL-trained representations.

Having proposed the SSL Model and its ELBO as a theoretical rationale behind self-supervised learning, we anticipate that directly maximising $\text{ELBO}_{\text{SSL}}$ (termed SimVAE) should give comparable or improved representations. However, it is well known that, at present, generative modelling brings additional challenges relative to discriminative modelling (He et al., 2022), making results of discriminative methods a benchmark to aspire to and VAE-based generative methods a relevant point of comparison. As such, we show that SimVAE representations i) consistently outperform other VAE-based representations on downstream tasks; ii) compare to or even outperform predictive SSL methods at downstream classification on simple benchmark datasets (MNIST, FashionMNIST and CelebA); iii) significantly bridge the gap between generative and discriminative methods for content classification on more complex datasets; and iv) consistent with our analysis, outperform discriminative methods on tasks that require style information.

Overall, our results provide empirical support for the SSL Model as a mathematical basis for self-supervised learning and suggest that SSL methods may overfit to content classification tasks. They also indicate that *generative* SSL may be a promising approach to task-agnostic representation learning if distributions can be well modelled, particularly given the potential added benefits of: uncertainty estimation from the posterior, a means to generate novel samples and qualitatively assess captured information from reconstructions, and to model arbitrary sets of semantically related data, not only pairs.

To summarise our main contributions:

- we propose the SSL Model (Fig. 1) and its ELBO as a theoretical basis to justify and unify predictive SSL, several families of self-supervised learning algorithms, including popular contrastive methods (§3);

- we show that the SSL Model predicts that discriminative methods lose *style* information and rationalises: the view that SSL methods "pull together"/"push apart" representations, a perceived link to mutual information, and the use of a projection head (§4); and

- we show that SimVAE representations, learned generatively by maximising $\text{ELBO}_{\text{SSL}}$, outperform predictive SSL methods at *content* prediction on simpler benchmarks and at tasks requiring *style* information across multiple benchmarks (+14.8% for CelebA); and significantly outperform previous generative (VAE-based) methods on content classification tasks (+15% on CIFAR10), including one tailored to SSL (§7).

## 2 Background and Related Work

**Representation Learning** aims to learn a mapping $f : \mathcal{X} \to \mathcal{Z}$ (an *encoder*) from data $x \in \mathcal{X}$ to representations $z = f(x) \in \mathcal{Z}$ (typically $|\mathcal{Z}| < |\mathcal{X}|$) that perform well on downstream tasks. Representation learning is not "well defined" in that downstream tasks are arbitrary and good performance on one may not mean good performance on another (Zhang et al., 2022). For instance, image representations are commonly evaluated by predicting semantic class labels, but the downstream task could instead be to detect lighting, position or orientation, which representations useful for predicting class may not capture. This suggests that *general-purpose* or *task-agnostic* representations should capture as much information about the data as possible, as supported by recent works that evaluate on a range of downstream tasks (e.g. Balažević et al., 2023).

**Self-Supervised Learning** includes many approaches that can be categorised in several ways (e.g. Balestriero et al., 2023; Garrido et al., 2022). The SSL methods that we focus on (**predictive SSL**) are defined below:

Instance Discrimination (Dosovitskiy et al., 2014; Wu et al., 2018) treats each data point $x_i$ and any of its augmentations as samples of a distinct class labelled by the index $i$. A softmax classifier (encoder + softmax layer) is trained to predict the "class" label (i.e. index) and encoder outputs are taken as representations.

Latent Clustering performs clustering on representations. Song et al. (2013); Xie et al. (2016); Yang et al. (2017) apply K-means, or similar, to the hidden layer of a standard auto-encoder. DeepCluster (Caron et al., 2020) iteratively clusters ResNet encoder outputs by K-means and uses cluster assignments as "pseudo-labels" to train a classifier. DINO (Caron et al., 2021), a transformer-based model, can be interpreted similarly as clustering representations (Balestriero et al., 2023).

Contrastive Learning encourages representations of semantically related data (positive samples) to be "close" relative to those sampled at random (negative samples). Early SSL approaches include energy-based models (Chopra et al., 2005; Hadsell et al., 2006); and word2vec (Mikolov et al., 2013) that predicts co-occurring words and encodes their pointwise mutual information (PMI) in its embeddings (Levy & Goldberg, 2014; Allen & Hospedales, 2019). InfoNCE (Oord et al., 2018; Sohn, 2016) extends word2vec to other data domains. For a positive pair of semantically related samples $(x, x^+)$ and randomly selected negative samples $X^- = \{x_k^-\}_{k=1}^K$, the InfoNCE objective is defined by:

$$\mathcal{L}_{\text{INCE}} \;=\; \mathbb{E}_{x, x^+, X^-} \left[ \log \frac{e^{\text{sim}(z, z^+)}}{\sum_{x' \in \{x^+\} \cup X^-} e^{\text{sim}(z, z')}} \right] , \tag{1}$$

where $\text{sim}(\cdot, \cdot)$ is a similarity function, e.g. dot product. The InfoNCE objective (Eq. 1) is optimised if $\text{sim}(z, z') = \text{PMI}(x, x') + c$, for some constant $c$ (Oord et al., 2018). Many works build on InfoNCE, e.g. SimCLR (Chen et al., 2020a) uses synthetic augmentations and CLIP (Radford et al., 2021) uses different modalities as positive samples; DIM (Hjelm et al., 2019) takes other encoder parameters as representations; and MoCo (He et al., 2020), BYOL (Grill et al., 2020) and VicREG (Bardes et al., 2022) find alternative strategies to negative sampling to prevent representations from collapsing.

Due to their wide variety, we do not address all SSL algorithms, in particular those with regression-style auxiliary tasks, e.g. reconstructing data from perturbed versions (He et al., 2022; Xie et al., 2022); or predicting perturbations, e.g. rotation angle (Gidaris et al., 2018).

**Variational Auto-Encoder (VAE).** For a generative latent variable model $z \to x$, parameters $\theta$ of a model $p_\theta(x) = \int_z p_\theta(x|z)p_\theta(z)$ can be learned by maximising the evidence lower bound (ELBO)

$$\mathbb{E}_x\big[\log p(x)\big] \quad \geq \quad \mathbb{E}_x\big[\log p_\theta(x)\big] \quad \geq \quad \mathbb{E}_x\left[\int_z q_\phi(z|x)\log\frac{p_\theta(x|z)p_\theta(z)}{q_\phi(z|x)}\right], \tag{ELBO}$$

where $q_\phi(z|x)$ learns to approximate the model posterior $p_\theta(z|x) \doteq \frac{p_\theta(x|z)p_\theta(z)}{\int_z p_\theta(x|z)p_\theta(z)}$. Latent variables $z$ can be used as representations (§4.1). A VAE (Kingma & Welling, 2014) maximises the ELBO, with $p_\theta$, $q_\phi$ modelled as Gaussians parameterised by neural networks. A $\beta$-VAE (Higgins et al., 2017) weights ELBO terms to increase disentanglement of latent factors. A CR-VAE (Sinha & Dieng, 2021) considers semantically related samples through an added regularisation term. Further VAE variants are summarised in Appendix A.1.

**Variational Classification (VC).** Dhuliawala et al. (2023) define a latent variable model for classifying labels $y$, $p(y|x) = \int_z q(z|x)p(y|z)$, that generalises softmax neural network classifiers. The encoder is interpreted as parameterising $q(z|x)$; and the softmax layer encodes $p(y|z)$ by Bayes' rule (**VC-A**). For continuous data domains $\mathcal{X}$, e.g. images, $q(z|x)$ of a softmax classifier (parameterised by the encoder) is shown to overfit to a delta-distribution for all samples of each class, and representations of a class "collapse" together (recently termed "neural collapse", Papyan et al., 2020), losing semantic and probabilistic information that distinguishes class samples and so harming properties such as calibration and robustness (**VC-B**).[2]

**Prior theoretical analysis of SSL.** There has been considerable interest in understanding the mathematical mechanism behind self-supervised learning (Arora et al., 2019; Tsai et al., 2020; Wang & Isola, 2020; Zimmermann et al., 2021; Lee et al., 2021; Von Kügelgen et al., 2021; HaoChen et al., 2021; Wang et al., 2021; Saunshi et al., 2022; Tian, 2022; Sansone & Manhaeve, 2022; Nakamura et al., 2023; Shwartz-Ziv et al., 2023; Ben-Shaul et al., 2023), as summarised by HaoChen et al. (2021) and Saunshi et al. (2022). A thread of works (Arora et al., 2019; Tosh et al., 2021; Lee et al., 2021; HaoChen et al., 2021; Saunshi et al., 2022) aims to prove that auxiliary task performance translates to downstream classification accuracy, but Saunshi et al. (2022) prove that to be impossible for typical datasets without also considering model architecture. Several works propose an information theoretic basis for SSL (Hjelm et al., 2019; Bachman et al., 2019; Tsai et al., 2020; Shwartz-Ziv et al., 2023), e.g. maximising mutual information between representations, but Tschannen et al. (2020); McAllester & Stratos (2020); Tosh et al. (2021) raise doubts about this. We show that this apparent connection to mutual information is justified more fundamentally by our model for SSL. The previous works most similar to our own are those that take a probabilistic view of self-supervised learning, in which the encoder approximates the posterior of a generative model and so can be interpreted to reverse the generative process (e.g. Zimmermann et al., 2021; Von Kügelgen et al., 2021; Daunhawer et al., 2023).

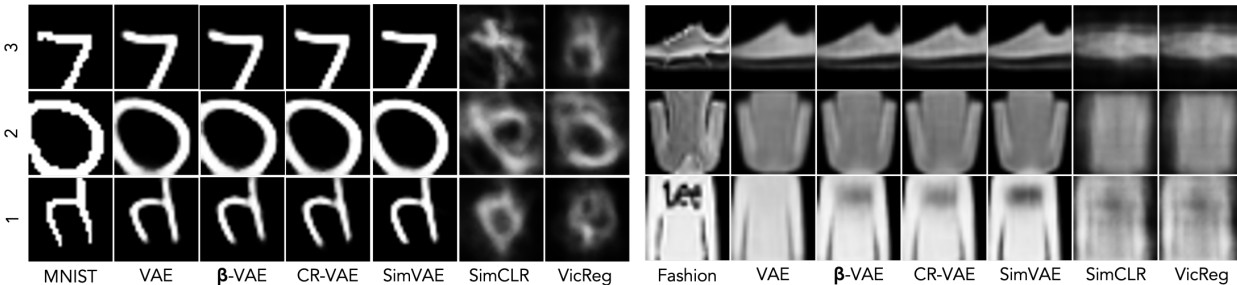

Figure 3: **Assessing the information in representations.** Original images (left cols) and reconstructions from representations learned by generative unsupervised learning (VAE, $\beta$-VAE, CR-VAE), generative SSL (our SimVAE) and discriminative SSL (SimCLR, VicREG) on MNIST ($l$), Fashion MNIST ($r$). Discriminative methods lose style information (e.g. orientation).

---

[2]In practice, constraints such as $l_2$ regularisation and early stopping arbitrarily restrict this optimum being fully attained.

# 3 A Probabilistic Model for Self-supervised Learning

Here, we propose a latent variable model for self-supervised learning, referred to as the SSL Model, and its evidence lower bound, ELBO$_{\text{SSL}}$, as the rationale behind predictive SSL methods.

We consider the data used for predictive SSL, $X = \bigcup_{i=1}^{N} \mathbf{x}_i$, as a collection of subsets $\mathbf{x}_i = \{x_i^j\}_j$ of *semantically related* data, where $x_i^j \in \mathcal{X}$ may, for example, be different augmentations, modalities or snippets (indexed $j$) of data observations $x_i$.[3] All $x_i^j \in \mathbf{x}_i$ are considered to have some semantic information in common, referred to as *content*, while varying in what we refer to as *style*, e.g. mirror images of an object reflect the same object (content) in different orientations (style). A hierarchical generative process for this data, the SSL Model, is shown in Fig. 1. Under this model, a subset $\mathbf{x}$ of $J$ semantically related data samples is generated by sampling: (i) $y \sim p(\mathrm{y})$, which determines the common semantic content, (ii) $z^j \sim p(\mathrm{z}|y)$, conditionally independently for $j \in [1, J]$, which determine the style of each $x^j$ (given the same $y$), and (iii) each $x^j \sim p(\mathrm{x}|z^j)$; hence

$$p(\mathbf{x}|y) = \int_{\mathbf{z}} \Big(\prod_{j=1}^{J} p(x^j|z^j)\Big)\Big(\prod_{j=1}^{J} p(z^j|y)\Big), \tag{2}$$

where $\mathbf{z} = \{z^j\}_{j=1}^{J}$ and $y$, which defines which $x$ are semantically related, is known, i.e. *observed*. If distributions in Eq. 2 are modelled parametrically, their parameters can be learned by maximising the evidence lower bound to the conditional log likelihood (analogous to the standard ELBO),

$$\mathop{\mathbb{E}}_{\mathbf{x},y}\big[\log p(\mathbf{x}|y)\big] \geq \mathop{\mathbb{E}}_{\mathbf{x},y}\left[\sum_{j=1}^{J}\int_{z^j} q_\phi(z^j|x^j)\big(\log p_\theta(x^j|z^j) - \log q_\phi(z^j|x^j)\big) + \int_{\mathbf{z}} q_\phi(\mathbf{z}|\mathbf{x})\log p_\psi(\mathbf{z}|y)\right], \quad (\textbf{ELBO}_{\textbf{SSL}})$$

where the approximate posterior $q(\mathbf{z}|\mathbf{x}) \approx \prod q(z^j|x^j)$ is assumed to factorise.[4] A derivation of **ELBO$_{\text{SSL}}$** is given in Appendix A.2. Similarly to the standard ELBO, ELBO$_{\text{SSL}}$ comprises: a reconstruction term, the approximate posterior entropy and the (now conditional) log prior over all $z^j \in \mathbf{z}$.

By assuming that $p(\mathrm{z}|y)$ are unimodal and concentrated relative to $p(\mathrm{z})$ (i.e. $\mathrm{Var}[\mathrm{z}|y] \ll \mathrm{Var}[\mathrm{z}], \forall y$), latent variables of semantically related data $z_i^j \in \mathbf{z}_i$ form clusters and are *closer on average than random pairs* – just as SSL representations are described. Thus, fitting the SSL Model to the data by maximising ELBO$_{\text{SSL}}$, induces a mixture prior distribution over latent variables, $p_{\textbf{SSL}(\mathbf{z})} = \sum_y p(\mathrm{z}|y)p(y)$ (Fig. 2, *right*), comparable to the distribution over representations induced by predictive SSL methods (Fig. 2, *left*). This connection goes to the heart of our main claim: that the SSL Model and ELBO$_{\text{SSL}}$ underpin predictive SSL algorithms (§2). To formalise this, we first establish how discriminative and generative methods relate so that the loss functions of discriminative predictive SSL methods can be compared to the generative ELBO$_{\text{SSL}}$ objective.

# 4 A Probabilistic Model behind Predictive SSL

So far, we have assumed that latent variables make useful representations without justification. We now justify this by considering the relationship between discriminative and generative representation learning (§4.1) in order to understand how the generative ELBO$_{\text{SSL}}$ objective can be emulated discriminatively (§4.2). We then review predictive SSL methods and show how they each emulate ELBO$_{\text{SSL}}$ (§4.3).

## 4.1 Discriminative vs Generative Representation Learning

Standard classification tasks can be approached *discriminatively* or *generatively*, defined by whether or not a distribution over the data space $\mathcal{X}$ is learned in the process. We make an analogous distinction for representation learning where, whatever the approach, the aim is to train an encoder $f : \mathcal{X} \to \mathcal{Z}$ so that representations $z = f(x)$ follow a distribution useful for downstream tasks, e.g. clustered by classes of interest.

Under a generative model $\mathrm{z} \to \mathrm{x}$, latent variables $z$ are assumed to determine semantic properties of the data $x$, which are predicted by the posterior $p(\mathrm{z}|x)$. Thus, learning an approximate posterior $q(z|x)$ (parameterised by

---

[3]$|\mathbf{x}_i|$ may vary and domains $\mathcal{X}^j \ni x_i^j$ can differ with modality.

[4]Expected to be reasonable for $z^j$ that carry high information about $x^j$, such that observing related $x^k$ or its representation $z^k$ provides negligible extra information, i.e. $p(z^j|x^j, x^k) \approx p(z^j|x^j)$.

$f$) by variational inference offers a generative route to learning *semantically meaningful* data representations. Meanwhile, the predictive SSL methods of interest train a deterministic encoder $f$ under a loss function without a distribution over $\mathcal{X}$ and are considered discriminative. We aim to reconcile these two approaches.

To discuss both approaches in common terms, we note that a deterministic encoder $f$ can be viewed as a (very low variance) posterior $p_f(z|x) \approx \delta_{z-f(x)}$, which, with $p(x)$, defines the joint distribution $p_f(x, z) = p_f(z|x)p(x)$ and the marginal over representations $p_f(z) \doteq \int_x p_f(x, z)$. Thus, in principle, an encoder can be trained so that representations follow a "useful" distribution $p^*(\mathrm{z})$ in one of two ways:

- *generatively*, by optimising the ELBO for a generative model with prior $p^*(z)$; or
- *discriminatively*, by optimising a loss function that is minimised if (and only if) $p_f(z) \approx p^*(z)$.

This defines a form of equivalence between discriminative and generative approaches allowing our claim to be restated as: predictive SSL methods induce a distribution $p_f(\mathrm{z})$ over representations comparable to the prior $p_{\mathrm{SSL}}(\mathrm{z})$ under the SSL Model.

## 4.2 Emulating a Generative Objective Discriminatively

Having seen in §4.1 that, in principle, an encoder can be trained generatively or discriminatively so that representations follow a target distribution $p^*(z)$, we now consider how this can be achieved in practice for the prior of the SSL Model, $p^* = p_{\mathrm{SSL}}$.

While $\mathrm{ELBO}_{\mathrm{SSL}}$ gives a principled generative objective to learn representations that fit $p_{\mathrm{SSL}}(\mathrm{z})$, a discriminative loss that is minimised when $p_f(\mathrm{z}) = p_{\mathrm{SSL}}(\mathrm{z})$ is unclear. Therefore, using $\mathrm{ELBO}_{\mathrm{SSL}}$ as a template, we consider the individual effect each term has on the optimal posterior $q(z|x)$ (parameterised by encoder $f$) to understand how a discriminative (or "$p(x|z)$-free") objective might induce a similar distribution over representations.

- **Entropy**: as noted in §4.1, discriminative methods can be considered to have a posterior with very low fixed variance, $q(z|x) \approx \delta_{z-f(x)}$, for which $\mathcal{H}[q]$ is constant and is omitted from a discriminative objective.

- **Prior**: $\mathbb{E}_q[\log p_\psi(\mathbf{z}|y)]$ is optimal w.r.t. $q$ *iff* all related samples $x^j \in \mathbf{x}$ map to modes of $p(\mathrm{z}|y)$. For unimodal $p(\mathrm{z}|y)$, this means representations $z^j \in \mathbf{z}$ *collapse* to a point, losing style information that distinguishes $x^j \in \mathbf{x}$.[5] Since the prior term is not generative it is included directly in a discriminative objective.

- **Reconstruction**: $\mathbb{E}_q[\log p_\theta(x^j|z^j)]$ terms are maximised w.r.t. $q$ *iff* each $x^j$ maps to a *distinct* representation $z^j$, which $p_\theta(\mathrm{x}|z^j)$ maps back to $x^j$; countering the prior to *prevent collapse*. In a discriminative objective, this generative term is excluded, and is considered *emulated* by a term that requires $z^j$ to be distinct.

We see that $\mathrm{ELBO}_{\mathrm{SSL}}$ includes a term acting to collapse clusters $p(\mathrm{z}|y)$ and another preventing such collapse, which counterbalance one another when the full $\mathrm{ELBO}_{\mathrm{SSL}}$ objective is maximised. Since the reconstruction term, $p(x|z)$, must be excluded from a discriminative objective, a discriminative objective $\ell_{\mathrm{SSL}}$ is considered to emulate $\mathrm{ELBO}_{\mathrm{SSL}}$ if it combines the prior term with a ($p(x|z)$-free) substitute for the reconstruction term, $\mathfrak{R}(\cdot)$, that prevents representation collapse (both between and within clusters), i.e.

$$\ell_{\mathrm{SSL}} = \mathbb{E}_{\mathbf{x},y}\left[\int_{\mathbf{z}} q(\mathbf{z}|\mathbf{x})(\log p(\mathbf{z}|y) + \mathfrak{R}(\mathbf{z}, \mathbf{x}))\right]. \tag{3}$$

We can now restate our claim as: predictive SSL objectives have the form of Eq. 3 and so emulate $\mathrm{ELBO}_{\mathrm{SSL}}$ to induce a distribution over representations comparable to the prior $p_{\mathrm{SSL}}(\mathrm{z})$ of the SSL Model (Fig. 1).

The analysis of ELBO terms shows that objectives of the form of Eq. 3 fit the view that SSL "pulls together" representations of related data and "pushes apart" others (e.g. Wang & Isola, 2020). That is, the prior "pulls" representations to modes and the $\mathfrak{R}(\cdot)$ term "pushes" to avoid collapse. Thus $\mathrm{ELBO}_{\mathrm{SSL}}$, which underpins Eq. 3, provides the underlying rationale for this "pull/push" perspective of SSL, depicted in Fig. 2.

Lastly, we note that the true reconstruction in $\mathrm{ELBO}_{\mathrm{SSL}}$ not only avoids collapse but is a necessary component in order to approximate the posterior and learn meaningful representations. We will see, as may be expected, that substituting this term by $\mathfrak{R}(\cdot)$ can impact representations and their downstream performance.

---

[5]This assumes classes $\mathbf{x}_i$ are distinct, as is the case for empirical self-supervised learning datasets of interest.

### 4.3 Emulating the SSL Model with Predictive Self-Supervised Learning

Having defined $\mathrm{ELBO}_{\mathrm{SSL}}$ as a generative objective for self-supervised representation learning and $\ell_{\mathrm{SSL}}$ as a discriminate approximation to it, we now consider classes of predictive SSL, specifically instance discrimination, latent clustering and contrastive methods, and their relationship to $\ell_{\mathrm{SSL}}$, and thus the SSL Model.

**Instance Discrimination (ID)** trains a softmax classifier on data samples labelled by their index $\{(x_i^j, y_i = i)\}_{i,j}$. From VC-A (§2), this softmax cross-entropy objective can be interpreted under the SSL Model as maximising a variational lower bound given by

$$\mathbb{E}_{x,y}[\log p(y|x)] \geq \mathbb{E}_{x,y}\left[\int_z q(z|x) \log p(y|z)\right] = \mathbb{E}_{x,y}\left[\int_z q(z|x)(\log p(z|y) - \log p(z))\right] + c. \quad (4)$$

Eq. 4 (RHS) matches Eq. 3 with $J=1$ and $\mathfrak{R}(\cdot) = \mathcal{H}[p(\mathrm{z})]$, the entropy of $p(\mathrm{z})$, where $z = f(x)$ are outputs of the classifier encoder. Intuitively, maximising entropy (in lieu of the reconstruction term) might be expected to avoid collapse, but although representations ($z$) of distinct classes are spread apart, those within the same class, $z_i^j \in \mathbf{z}_i$, collapse together (VC-B, §2).

**Deep Clustering (DC)** iteratively assigns temporary labels $y = c$, $c \in [1, C]$, to data $x$ by clustering representations $z$ output by a ResNet encoder; and trains the encoder together with a softmax head to predict those labels. While subsets $\mathbf{x}$ of data with the same label are now defined according to a ResNet's "inductive bias", the same loss is used as Eq. 4, having the form of Eq. 3 with $J=1$. Each cluster of representations $z^j \in \mathbf{z}$ again collapses together.

**Contrastive Learning** based on the InfoNCE objective (Eq. 1), contrasts the representations of positive pairs $\mathbf{x} = \{x, x^+\}$ with those sampled at random. The InfoNCE objective is known to be optimised when $\mathrm{sim}(z, z') = \mathrm{PMI}(x, x') + c$ (§2). From this, it can be shown that for $X' = \{x^+, x_1^-, ..., x_k^-\}$, $X = X' \cup \{x\}$,

$$\mathcal{L}_{\mathrm{INCE}}(x, X') = \mathbb{E}_X\left[\int_z q_\phi(Z|X) \log \frac{\mathrm{sim}(z, z^+)}{\sum_{z' \in Z'} \mathrm{sim}(z, z')}\right] \leq \mathbb{E}_X\left[\int_Z q_\phi(Z|X) \log \frac{p(z^+|z)/p(z^+)}{\sum_{z' \in Z'} p(z'|z)/p(z')}\right]$$

$$\lesssim \mathbb{E}_{\mathbf{x}}\left[\int_{\mathbf{z}} q_\phi(\mathbf{z}|\mathbf{x}) \log \frac{p(z, z^+)}{p(z)p(z^+)}\right] - \log(k-1)$$

$$= \mathbb{E}_{\mathbf{x}}\left[\int_{\mathbf{z}} q_\phi(\mathbf{z}|\mathbf{x})(\log p(\mathbf{z}|y) - \sum_j \log p(z^j))\right] - c \quad (5)$$

(See Appendix A.3 for a full derivation.) Here, $p(z, z^+)$ is the probability of sampling $z, z^+$ assuming that $x, x^+$ are semantically related. Under the SSL Model this is denoted $p(z, z^+|y) \doteq p(\mathbf{z}|y)$, hence the change in notation for ease of comparison to Eq. 3. Thus, the InfoNCE loss lower bounds an objective of the form of Eq. 3 with $\mathfrak{R}(\cdot) = \mathcal{H}[p(z)]$ as in Eq. 4, but with $J=2$ samples.

As a brief aside, we note that ID and DC consider $J=1$ sample $x_i^j$ at a time, computing $p(z_i|y_i; \psi_i)$ from parameters $\psi_i$ stored for each class (weights of the softmax layer), which could be memory intensive. However, for $J \geq 2$ samples, one can estimate $p(\mathbf{z}_i|y_i)$ without stored parameters as $p(\mathbf{z}_i|y_i) = \int_{\psi_i} p(\psi_i) \prod_j p(z_i^j|y_i; \psi_i) \doteq s(\mathbf{z}_i)$ if $\psi_i$ can be integrated out (see Appendix A.4 for an example). Under this "$\psi$-free" approach, joint distributions over representations depend on whether they are semantically related:

$$p(z_{i_1}, ..., z_{i_k}) = \begin{cases} s(\mathbf{z}_i) & \text{... if } i_r = i \;\; \forall r \text{ (i.e. } x_{i_r} \in \mathbf{x}_i) \\ \prod_{r=1}^k p(z_{i_r}) & \text{... if } i_r \neq i_s \, \forall r, s \end{cases}$$

InfoNCE implicitly employs this "trick" within $\mathrm{sim}(\cdot, \cdot)$ to avoid parameterisation of each cluster.

To summarise, we have considered the loss functions of each category of predictive SSL and shown that:

1. despite their differences, each has the form of $\ell_{\mathrm{SSL}}$ and so emulates $\mathrm{ELBO}_{\mathrm{SSL}}$ to induce a distribution over representations comparable to the prior of the SSL Model; and

2. each method substitutes the reconstruction term of $\mathrm{ELBO}_{\mathrm{SSL}}$ by entropy $\mathcal{H}[p(z)]$, which causes representations of semantically related data $z_i^j \in \mathbf{z}_i$ to form distinct clusters (indexed by $i$), as in $p_{\mathrm{SSL}}$, but those clusters collapse and lose style information as representations (indexed by $j$) become indistinguishable.

Style information is important in many real-world tasks, e.g. detecting an object's location or orientation; and is the main focus of representation learning elsewhere (Higgins et al., 2017; Karras et al., 2019). Thus, contrary to the aim of *general* representation learning, our analysis predicts that **discriminative SSL over-fit to content-related tasks** (discussed further in Appendix A.6).

Even restricting to predictive SSL, an exhaustive analysis of methods is infeasible, however, many other methods adopt or approximate aspects analysed above. For example, SimCLR (Chen et al., 2020a) and CLIP (Radford et al., 2021) use the InfoNCE objective. MoCo (He et al., 2020), BYOL (Grill et al., 2020) and VicREG (Bardes et al., 2022) replace negative sampling in different ways but ultimately "pull together" representations of semantically related data, modelling the prior; and "push apart" others, avoiding collapse via a mechanism (e.g. momentum encoder, stop gradients or (co-)variance terms) in lieu of reconstruction. DINO (Caron et al., 2021) assigns representations of pairs of semantically related data to the same cluster, as in DC (Balestriero et al., 2023) but with $J = 2$.

Lastly, we discuss how the SSL Model also provides a principled explanation for the connection between SSL and mutual information drawn in prior works, and a plausible rationale for using a projection head in SSL.

### 4.3.1 Relationship to Mutual Information

InfoNCE is known to optimise a lower bound on *mutual information* (MI), $I(x, x') = \mathbb{E}[\log \frac{p(x, x')}{p(x)p(x')}]$, (Oord et al., 2018) and Eq. 5 shows it lower bounds $I(z, z')$. It has been argued that maximising MI is in fact the underlying rationale that explains contrastive learning (Hjelm et al., 2019; Ozsoy et al., 2022). Previous works have challenged this showing that better MI estimators do not give better representations (Tschannen et al., 2020), that MI approximation is inherently noisy (McAllester & Stratos, 2020) and that the InfoNCE estimator is arbitrarily upper-bounded (Poole et al., 2019). Note also that for disjoint $\mathbf{x}_i$ (e.g. non-overlapping augmentation sets), the range of pointwise mutual information values is *unbounded*, $[-\infty, k]$, $k > 1$, yet using the *bounded* cosine similarity function, $\text{sim}(z, z') = \frac{z^\top z'}{\|z\|\|z'\|} \in [-1, 1]$ (e.g. Chen et al., 2020a) to model PMI values outperforms use of the unbounded dot-product (see §A.5).[6] Our analysis supports the idea that MI is *not* the fundamental mechanism behind SSL and explains the apparent connection: MI arises by substituting entropy $\mathcal{H}[p(x)]$ for the reconstruction term in $\text{ELBO}_{\text{SSL}}$ as in predictive SSL methods. Rather than aide representation learning, this is shown to collapse representations together and lose information.

### 4.3.2 Rationale for a Projection Head

Downstream performance is often found to improve by adding several layers to the encoder, termed a "projection head", and using encoder outputs as representations and projection head outputs in the loss function. Our analysis has shown that the $\text{ELBO}_{\text{SSL}}$ objective clusters representations of semantically related data while ensuring that all representations are distinct, whereas predictive SSL objectives collapse those clusters. However, near-final layers are found to exhibit similar clustering, but with higher intra-cluster variance (Gupta et al., 2022), as also observed in supervised classification (Wang et al., 2022). We conjecture that representations from near-final layers outperform those used in the loss function because their higher intra-cluster variance, or lower collapse, gives a distribution closer to $p_{\text{SSL}}$ of the SSL Model.

## 5 Generative Self-Supervised Learning (SimVAE)

The proposed SSL Model (Fig. 1) has been shown to justify: (i) the training objectives of predictive SSL methods; (ii) the more general notion that SSL "pulls together"/"pushes apart" representations; (iii) the connection to mutual information; and (iv) the use of a projection head.

As a proposed basis for self-supervised learning, we look to provide empirical validation by maximising $\text{ELBO}_{\text{SSL}}$ to (generatively) learn representations that perform comparably to predictive SSL. Maximising $\text{ELBO}_{\text{SSL}}$ can be viewed as training a VAE with mixture prior $p_{\text{SSL}}(\mathbf{z}) = \sum_y p(\mathbf{z}|y)p(y)$ where representations $z \in \mathbf{z}$ of semantically related data are conditioned on the same $y$, which we refer to as **SimVAE**.

---

[6]In §A.5, we show that cosine similarity gives representations comparable to using softmax cross entropy (ID) but without stored class parameters $\psi_i$ for each $x_i$. Wang & Isola (2020) note this conclusion but do not consider the known PMI minimiser.

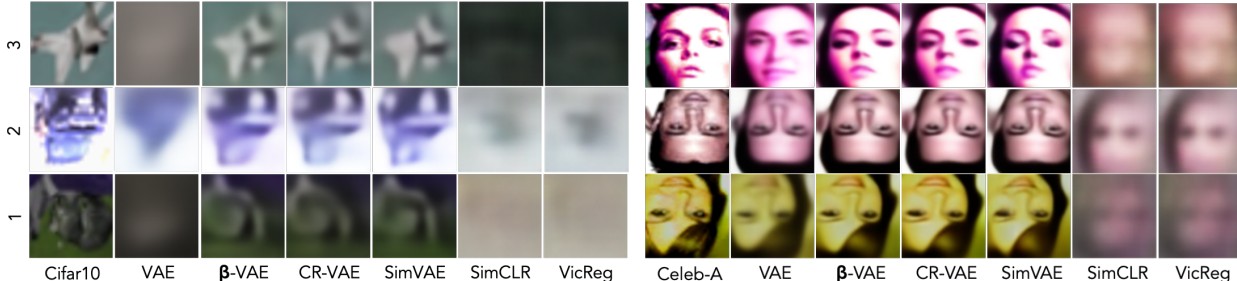

Figure 4: **Assessing information in representations.** Original images (left cols) and reconstructions from representations learned by generative unsupervised learning (VAE, $\beta$-VAE, CR-VAE), generative SSL (our SimVAE) and discriminative SSL (SimCLR, VicREG) on Cifar10 ($l$), CelebA ($r$). Discriminative methods lose style information (e.g. orientation/colour).

In practice, it is well known that training a generative model is more challenging, at present, than training a discriminative model (Radford et al., 2021; Tschannen et al., 2024), not least because it requires modelling a complex distribution in the high dimensional data space. SimVAE performance is therefore anticipated to be most comparable to other VAE-based approaches. Thus we add empirical support for the SSL Model by validating the following hypotheses: [**H1**] SimVAE achieves self-supervised learning *if* distributions can be well modelled; [**H2**] SimVAE retains more style information than discriminative objectives; [**H3**] SimVAE learns better performing representations than other VAE-based objectives.

**Implementing SimVAE.** Testing hypotheses [H1-3] requires instantiating distributions in $\text{ELBO}_{\text{SSL}}$. As for a standard VAE, we assume $p(x|z)$ and $q(z|x)$ are Gaussians parameterized by neural networks. The prior $p(y)$ is assumed uniform over $\{1, \ldots, N\}$, for N training samples and $p(z \mid y=i; \psi_i) = \mathcal{N}(z; \psi_i, \sigma^2)$ are Gaussian with small fixed variance $\sigma^2$. Assuming a uniform distribution $p(\psi)$ over their means (over a suitable space), $\psi$ integrates out (see Appendix A.4) to give:

$$p(\mathbf{z}|y) \;\propto\; \exp\{-\tfrac{1}{2\sigma^2} \sum_j (z^j - \bar{z})^2\}, \tag{6}$$

a Gaussian centred on the mean representation $\bar{z} = \frac{1}{J} \sum_j z^j$. Maximising this within $\text{ELBO}_{\text{SSL}}$ can be considered to "pinch together" representations of semantically related data.

While contrastive methods typically compare *pairs* of related representations ($J=2$), $\text{ELBO}_{\text{SSL}}$ allows any number $J$ to be compared. In practice a balance is struck between better approximating $p(\mathbf{z}|y)$ (high $J$) and diversity in a mini-batch (low $J$). Algorithm 1 in Appendix A.7.1 details the steps to optimise $\text{ELBO}_{\text{SSL}}$.

# 6 Experimental Setup

**Datasets and Evaluation Metrics.** We evaluate SimVAE representations on four datasets including two with natural images: MNIST (LeCun, 1998), FashionMNIST (Xiao et al., 2017), CelebA (Liu et al., 2015) and CIFAR10 (Krizhevsky et al., 2009). We augment images following the SimCLR protocol (Chen et al., 2020a) which includes cropping and flipping, and colour jitter for natural images. Frozen pre-trained representations are evaluated by a $k$-Nearest Neighbors ($k$-NN; Cover & Hart, 1967)($k$-NN) and a non-linear MLP probe on classification tasks (Chen et al., 2020a; Caron et al., 2020). Downstream performance is measured in terms of classification accuracy (Acc). Generative quality is evaluated by FID score (Heusel et al., 2017) and reconstruction error. For further experimental details and additional results for clustering under a gaussian mixture model and a linear probe see Appendices A.7 and A.8.

**Baselines methods.** We compare SimVAE to other VAE-based models including the vanilla VAE (Kingma & Welling, 2014), $\beta$-VAE (Higgins et al., 2017) and CR-VAE (Sinha & Dieng, 2021), as well as to state-of-the-art self-supervised discriminative methods including SimCLR (Chen et al., 2020a), VicREG (Bardes et al., 2022), MoCo (He et al., 2020) and its extension MoCo v2 (Chen et al., 2020b). As a lower bound,

| | Acc-MP | | | | Acc-$k$-NN | | | |
|---|---|---|---|---|---|---|---|---|
| | MNIST | Fashion | CelebA | CIFAR10 | MNIST | Fashion | CelebA | CIFAR10 |
| Random | 38.1 ± 3.8 | 49.8 ± 0.8 | 83.5 ± 1.0 | 16.3 ± 0.4 | 46.1 ± 2.5 | 66.5 ± 0.4 | 80.0 ± 0.9 | 13.1 ± 0.6 |
| SimCLR | **97.2** ± 0.0 | **74.9** ± 0.2 | **93.7** ± 0.4 | 67.4 ± 0.1 | **97.2** ± 0.1 | 76.0 ± 0.1 | 91.6 ± 0.3 | 64.0 ± 0.0 |
| MoCo | 78.6 ± 1.2 | 65.1 ± 0.4 | 91.2 ± 0.1 | 56.4 ± 1.6 | 94.6 ± 0.3 | **76.9** ± 0.2 | 87.9 ± 0.1 | 54.0 ± 2.0 |
| MoCov2 | 94.6 ± 0.4 | 71.2 ± 0.1 | 91.7 ± 0.1 | 56.4 ± 1.6 | 94.6 ± 0.3 | **76.9** ± 0.2 | 88.8 ± 0.4 | 54.0 ± 2.0 |
| VicREG | 96.7 ± 0.0 | 73.2 ± 0.1 | 94.7 ± 0.1 | **69.7** ± 0.0 | 97.0 ± 0.0 | 76.0 ± 0.1 | **92.7** ± 0.4 | **68.3** ± 0.0 |
| VAE | 97.8 ± 0.1 | 80.2 ± 0.3 | 89.0 ± 0.5 | 30.3 ± 0.4 | 98.0 ± 0.1 | 83.7 ± 0.2 | 86.9 ± 0.7 | 25.6 ± 0.5 |
| $\beta$-VAE | 98.0 ± 0.0 | 82.2 ± 0.1 | 93.4 ± 0.4 | 36.6 ± 0.1 | 98.3 ± 0.0 | 86.1 ± 0.0 | 92.0 ± 0.1 | 28.5 ± 0.1 |
| CR-VAE | 97.7 ± 0.0 | **82.6** ± 0.0 | 93.1 ± 0.4 | 36.8 ± 0.0 | 98.0 ± 0.0 | 86.4 ± 0.0 | 91.6 ± 0.6 | 28.1 ± 0.1 |
| SimVAE | **98.4** ± 0.0 | 82.1 ± 0.0 | **95.6** ± 0.4 | **51.8** ± 0.0 | **98.5** ± 0.0 | **86.5** ± 0.0 | **93.2** ± 0.1 | **47.1** ± 0.0 |

Table 1: **Content retrieval.** Top-1% classification accuracy($\uparrow$) for MNIST, FashionMNIST, CIFAR10, and CelebA (gender classification) using a MLP probe (MP) and $k$-Nearest Neighbors ($k$-NN) classification methods; We report mean and standard errors over three runs; Bold indicates best scores in each method class: generative (teal), discriminative methods (red).

we also provide results for a randomly initialized encoder. For fair comparison, the augmentation strategy, representation dimensionality, batch size, and encoder-decoder architectures are invariant across methods. To enable a qualitative comparison of representations, decoder networks were trained for each discriminative baseline on top of frozen representations using the reconstruction error. See Appendix A.7 for further details on training baselines and decoder training.

**Implementation Details.** We use MLP and Resnet18 (He et al., 2016) network architectures for simple and natural image datasets respectively. For all generative approaches, we adopt Gaussian posteriors, $q(z|x)$, priors, $p(z)$, and likelihoods, $p(x|z)$, with diagonal covariance matrices (Kingma & Welling, 2014). For SimVAE, we adopt Gaussian $p(z|y)$ as described in §5 and, we fix the number of augmentations to $J=10$ (see Fig. 8 for an ablation). Ablations were performed for all sensitive hyperparameters for each method and parameter values were selected based on the best average MLP Acc across datasets. Further details regarding hyperparameters and computational resources can be found in Appendix A.7.

## 7 Results

We assess content and style information retrieval by predicting attributes that are preserved (e.g. object class, face gender) or disrupted (e.g. colour, position and orientation) under augmentation, respectively.

**Content Retrieval.** Table 1 reports downstream classification performance across datasets using class labels. SimVAE outperforms predictive SSL for simple datasets (MNIST, FashionMNIST, CelebA), which empirically supports **H1**, demonstrating that SimVAE achieves self-supervised learning where data distributions can be well-modelled. SimVAE also materially reduces the performance gap ($\Delta$) between generative and discriminative SSL, by approximately half ($\Delta = 32.8\% \rightarrow 17.6\%$) for more complex settings (CIFAR10). The performance improvement by SimVAE over other generative methods supports **H3**.

**Style Retrieval.** We reconstruct MNIST, FashionMNIST, CIFAR-10, and CelebA images from representations to gain qualitative insights into the style information captured. Figures 3 and 4 show a significant loss of orientation and colour in reconstructions from predictive SSL representations across datasets. CelebA has 40 mixed labeled attributes, some of which clearly pertain to style (e.g. hair color). Removing those with high-class imbalance, we evaluate classification of 20 attributes and report results in Figure 5. SimVAE outperforms both generative and predictive SSL baselines at classifying hair color (*left, middle*). Across all attributes (*right*), SimVAE outperforms or performs comparably to both generative and predictive SSL baselines. These results support **H2** qualitatively and quantitatively, confirming that discriminative methods lose more style information relative to generative methods (Figure 5, *middle*). Performance for each CelebA attribute is reported in Figure 6 in Appendix A.8.

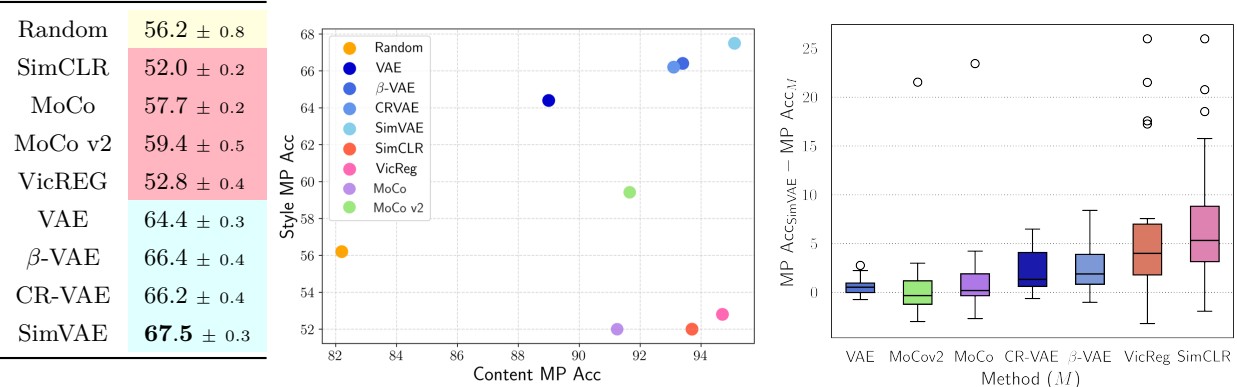

Figure 5: **Style retrieval on CelebA.** (Acc-MP ↑) (*left*) hair colour prediction (mean and standard error over 3 runs, best results in bold); (*middle*) content vs. style prediction (gender vs hair colour), best performance in top-right; (*right*) Performance gain of SimVAE vs baselines (M) across all 20 CelebA attributes. From left to right, lowest to highest *mean* score;

**Image Generation.**   While generative quality is not relevant to our main hypotheses, out of interest we show randomly generated SimVAE images and quality metrics in Appendix A.8. We observe small but significant improvements in FID score and reconstruction error relative to previous VAE methods.

## 8   Conclusion

The impressive performance of self-supervised learning over recent years has spurred considerable interest in understanding the theoretical mechanism underlying SSL. In this work, we propose a latent variable model (SSL Model, Fig. 1) as the theoretical basis to explain several families of self-supervised learning methods, termed "predictive SSL", including contrastive learning. We show that the ELBO of the SSL Model relates to the loss functions of predictive SSL methods and justifies the general notion that SSL "pulls together" representations of semantically related data and "pushes apart" others.

Predictive SSL methods are found to maximise entropy $\mathcal{H}[p(z)]$, in lieu of the ELBO's reconstruction term. Although this creates a seemingly intuitive link to the *mutual information* between representations that was thought to explain contrastive methods, it in fact causes representations of semantically related data to collapse together, losing *style* information about the data, and reducing their generality. Our analysis also justifies the use of a "projection head", the existence of which suggests caution in (over-)analysing representations that are ultimately not used.

We provide empirical validation for the proposed SSL Model by showing that fitting it by variational inference, termed SimVAE, learns representations generatively that perform comparably, or significantly reduce the performance gap, to discriminative methods at *content* prediction. Meanwhile, SimVAE outperforms where *style* information is required, taking a step towards task-agnostic representations. SimVAE also outperforms previous generative VAE-based approaches, including CR-VAE tailored to SSL.

Learning representations of complex data distributions generatively, rather than discriminatively, remains a challenging and actively researched area (Tschannen et al., 2024). Balestriero & LeCun (2024) recently highlighted the scale of the challenge for higher-variance datasets such as ImageNet (Deng et al., 2009). We hope that the theoretical connection to popular SSL methods and the notable performance improvements of SimVAE over other VAE-based methods encourages further investigation into generative representation learning. This seems particularly justified given the potential for uncertainty estimation from the posterior, the ability to generate novel data samples, and the potential for fully task-agnostic representations learned under a principled rather than heuristic approach. The transparency of the SSL Model may also allow representations to be learned that disentangle rather than discard style information (Higgins et al., 2017) through careful choice of model parameters. The SSL Model thus serves as a basis for understanding self-supervised learning and offers guidance for the design and interpretation of future representation learning methods.

**Acknowledgments**

Alice Bizeul is gratefully supported by an ETH AI Center Doctoral Fellowship and by Professors Julia Vogt and Bernhard Schölkopf. Carl is gratefully supported by an ETH AI Centre Postdoctoral Fellowship, a grant from the Haslerstiftung (no. 23072) and by Professors Mrinmaya Sachan and Ryan Cotterell. We thank Pawel Czyz and Thomas Sutter for their valuable feedback.

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

# A   Appendix

## A.1   Background: Relevant VAE architectures

The proposed hierarchical latent variable model for self-supervised learning (Fig. 1) is fitted to the data distribution by maximising the ELBO$_{\text{SSL}}$ and can be viewed as a VAE with a hierarchical prior.

VAEs have been extended to model hierarchical latent structure (e.g. Valpola, 2015; Ranganath et al., 2016; Rolfe, 2017; He et al., 2018; Sønderby et al., 2016; Edwards & Storkey, 2016), which our work relates to. Notably, Edwards & Storkey (2016) propose the same graphical model as Fig. 1, but methods differ in how posteriors are factorised, which is a key aspect for learning informative representations that depend only on the sample they represent. Wu et al. (2023) and Sinha et al. (2021) combine aspects of VAEs and contrastive learning but do not propose a latent variable model for SSL. Nakamura et al. (2023) look to explain SSL methods via the ELBO, but with a second posterior approximation not a generative model.

## A.2   Derivation of ELBO$_{\text{SSL}}$ and SimVAE Objective

Let $\mathbf{x} = \{x^1, ..., x^J\}$ be a set of $J$ semantically related samples with common label $y$ (e.g. the index $i$ of a set of augmentations of an image $x_i$). Let $\omega = \{\theta, \psi, \pi\}$ be parameters of the generative model for SSL (Fig. 1) and $\phi$ be the parameter of the approximate posterior $q_\phi(\mathbf{z}|\mathbf{x})$. We derive the Evidence Lower Bound (ELBO$_{\text{SSL}}$) that underpins the training objectives of (projective) discriminative SSL methods and is used to train SimVAE (§5).

$$
\begin{aligned}
\min_\omega D_{\text{KL}}[\, p(\mathbf{x}|y) \,\|\, p_\omega(\mathbf{x}|y)\,] &= \max_\omega \mathop{\mathbb{E}}_{\mathbf{x},y}\left[\log p_\omega(\mathbf{x}|y)\right] \\
&= \max_{\omega,\phi} \mathop{\mathbb{E}}_{\mathbf{x},y}\left[\int_{\mathbf{z}} q_\phi(\mathbf{z}|\mathbf{x}) \log p_\omega(\mathbf{x}|y)\right] \\
&= \max_{\omega,\phi} \mathop{\mathbb{E}}_{\mathbf{x},y}\left[\int_{\mathbf{z}} q_\phi(\mathbf{z}|\mathbf{x}) \log \tfrac{p_\theta(\mathbf{x}|\mathbf{z})p_\psi(\mathbf{z}|y)}{p_\omega(\mathbf{z}|\mathbf{x},y)}\tfrac{q_\phi(\mathbf{z}|\mathbf{x})}{q_\phi(\mathbf{z}|\mathbf{x})}\right] \\
&= \max_{\omega,\phi} \mathop{\mathbb{E}}_{\mathbf{x},y}\left[\int_{\mathbf{z}} q_\phi(\mathbf{z}|\mathbf{x}) \log \tfrac{p_\theta(\mathbf{x}|\mathbf{z})p_\psi(\mathbf{z})|y}{q_\phi(\mathbf{z}|\mathbf{x})}\right] + D_{\text{KL}}[\, q_\phi(\mathbf{z}|\mathbf{x}) \,\|\, p_\omega(\mathbf{z}|\mathbf{x},y)\,] \\
&\geq \max_{\omega,\phi} \mathop{\mathbb{E}}_{\mathbf{x},y}\left[\int_{\mathbf{z}} q_\phi(\mathbf{z}|\mathbf{x})\Big\{\log \tfrac{p_\theta(\mathbf{x}|\mathbf{z})}{q_\phi(\mathbf{z}|\mathbf{x})} + \log p_\psi(\mathbf{z}|y)\Big\}\right] \quad\quad (cf \text{ Eq. ELBO}) \\
&= \max_{\omega,\phi} \mathop{\mathbb{E}}_{\mathbf{x},y}\left[\sum_j \Big\{\int_{z^j} q_\phi(z^j|x^j) \log \tfrac{p_\theta(x^j|z^j)}{q_\phi(z^j|x^j)}\Big\} + \int_{\mathbf{z}} q_\phi(\mathbf{z}|\mathbf{x}) \log p_\psi(\mathbf{z}|y)\right] \\
&\hspace{11cm} (= \text{ELBO}_{\text{SSL}})
\end{aligned}
$$

Terms of ELBO$_{\text{SSL}}$ are analogous to those of the standard ELBO: *reconstruction error*, *entropy* of the approximate posterior $H(q_\phi(z|\mathbf{x}))$ and the (conditional) prior. Algorithm 1 provides an overview of the computational steps required to maximise ELBO$_{\text{SSL}}$ under Gaussian assumptions described in §5, referred to as SimVAE. As our experimental setting considers augmentations as semantically related samples, Algorithm 1 incorporates a preliminary step to augment data samples.

## A.3   Detailed derivation of InfoNCE Objective

For data sample $x_{i_0} \in \mathbf{x}_{i_0}$, let $x'_{i_0} \in \mathbf{x}_{i_0}$ be a semantically related *positive* sample and $\{x'_{i_r}\}_{r=1}^k$ be random *negative* samples. Denote by $\mathbf{x} = \{x_{i_0}, x'_{i_0}\}$ the positive pair, by $X^- = \{x'_{i_1}, ..., x'_{i_k}\}$ all negative samples, and by $X = \mathbf{x} \cup X^-$ all samples. The InfoNCE objective is derived as follows (by analogy to Oord et al. (2018)).

$$
\begin{aligned}
&\mathbb{E}_X\left[\log p(y=0|X)\right] &&(\text{predict } y = \text{index of positive sample in } X^-) \\
&= \mathbb{E}_X\left[\log \int_Z p(y=0|Z)q(Z|X)\right] &&(\text{introduce latent variables } Z: Y \to Z \to X) \\
&\geq \mathbb{E}_X\left[\int_Z q(Z|X) \log p(y=0|Z)\right] &&(\text{by Jensen's inequality})
\end{aligned}
$$

$$= \mathbb{E}_X \Big[ \int_Z q(Z|X) \log \frac{p(Z'|z_{i_0}, y=0)\,\cancel{p(y=0)}}{\sum_{r=0}^k p(Z'|z_{i_0}, y=r)\,\cancel{p(y=r)}} \Big] \qquad \text{(Bayes rule, note } p(y\!=\!r)\!=\!\tfrac{1}{k},\ \forall r)$$

$$= \mathbb{E}_X \Big[ \int_Z q(Z|X) \log \frac{p(z'_{i_0}|z_{i_0}) \prod_{s \neq 0} p(z'_{i_s})}{\sum_{r=0}^k p(z'_{i_r}|z_{i_0}) \prod_{s \neq r} p(z'_{i_s})} \Big] \qquad \text{(from sample similarity/independence)}$$

$$= \mathbb{E}_X \Big[ \int_Z q(Z|X) \log \frac{p(z'_{i_0}|z_{i_0})/p(z'_{i_0})}{\sum_{r=0}^k p(z'_{i_r}|z_{i_0})/p(z'_{i_r})} \Big] \qquad \text{(divide through by } \prod_{s=0}^k p(z'_{i_s}))$$

$$= \mathbb{E}_X \Big[ \int_Z q(Z|X) \log \frac{p(z'_{i_0}, z_{i_0})/p(z'_{i_0})}{\sum_{r=0}^k p(z'_{i_r}, z_{i_0})/p(z'_{i_r})} \Big] \qquad (7)$$

The final expression is parameterised using a similarity function $\mathrm{sim}(z, z')$ to give the objective.

$$-\mathcal{L}_{\mathrm{INCE}} \doteq \mathbb{E}_X \Big[ \int_Z q(Z|X) \log \frac{\mathrm{sim}(z'_{i_0}, z_{i_0})}{\sum_{r=0}^k \mathrm{sim}(z'_{i_r}, z_{i_0})} \Big]$$

Oord et al. (2018) show, and Poole et al. (2019) confirm, that this loss is a lower bound on the mutual information, which improves with the number of negative samples $k$.

$$-\mathcal{L}_{\mathrm{INCE}}$$
$$= \mathbb{E}_X \Big[ \int_Z q(Z|X) \log \frac{p(z'_{i_0}, z_{i_0})/p(z'_{i_0})}{\sum_{r=0}^k p(z'_{i_r}, z_{i_0})/p(z'_{i_r})} \Big] \qquad \text{(multiply Eq. 7 through by } p(z_{i_0}))$$

$$= \mathbb{E}_X \Big[ \int_Z q(Z|X) \log p(z_{i_0}|z'_{i_0}) - \log \big( p(z_{i_0}|z'_{i_0}) + \sum_{r=1}^k p(z_{i_0}|z'_{i_r}) \big) \Big]$$

$$= -\mathbb{E}_X \Big[ \int_Z q(Z|X) \log \big( 1 + \sum_{r=1}^k \frac{p(z_{i_0}|z'_{i_r})}{p(z_{i_0}|z'_{i_0})} \big) \Big] \qquad \text{(divide through by } p(z_{i_0}|z'_{i_0}))$$

$$\approx -\mathbb{E}_{\mathbf{x}} \Big[ \int_{\mathbf{z}} q(\mathbf{z}|\mathbf{x}) \log \big( 1 + (k-1) \mathbb{E}_{x'_j} \big[ \int_{z'_j} q(z'_j|x'_j) \frac{p(z_{i_0}|z'_j)}{p(z_{i_0}|z'_{i_0})} \big) \big] \Big]$$

$$= -\mathbb{E}_{\mathbf{x}} \Big[ \int_{\mathbf{z}} q(\mathbf{z}|\mathbf{x}) \log \big( 1 + (k-1) \frac{p(z_{i_0})}{p(z_{i_0}|z'_{i_0})} \big) \Big]$$

$$\leq -\mathbb{E}_{\mathbf{x}} \Big[ \int_{\mathbf{z}} q(\mathbf{z}|\mathbf{x}) \log(k-1) \frac{p(z_{i_0})}{p(z_{i_0}|z'_{i_0})} \Big]$$

$$= \mathbb{E}_{\mathbf{x}} \Big[ \int_{\mathbf{z}} q(\mathbf{z}|\mathbf{x}) \log \frac{p(z_{i_0}, z'_{i_0})}{p(z_{i_0})p(z'_{i_0})} \Big] + \log \frac{1}{(k-1)}$$

$$= \mathbb{E}_{\mathbf{x}} \Big[ \int_z q_\phi(\mathbf{z}|\mathbf{x}) \big( \log p(\mathbf{z}|y_{i_0}) - \sum_j \log p(z^j_{i_0}) \big) \Big] + c$$

In the last step, we revert to the terminology used in the main paper for ease of reference.

### A.4 Derivation of parameter-free $p(\mathbf{z}_i|y_i) = s(\mathbf{z}_i)$

Instance Discrimination methods consider $J\!=\!1$ sample $x_i$ at a time, labelled by its index $y_i\!=\!i$, and computes $p(x_i|\mathrm{y} = i; \theta_i)$ from *stored* instance-specific parameters $\theta_i$. This requires parameters proportional to the dataset size, which could be prohibitive, whereas parameter number is often independent of the dataset size, or grows slowly. We show that contrastive methods (approximately) optimise the same objective, but without parameters, and here explain how that is possible. Recall that the "label" $i$ is semantically meaningless and simply identifies samples of a common distribution $p(\mathrm{x}|\mathrm{y}\!=\!i) \doteq p(\mathrm{x}|y_i)$. For $J\!\geq\!2$ semantically related samples $\mathbf{x}_i = \{x_i^j\}_{j=1}^J$, $x_i^j \sim p(\mathrm{x}|y_i)$, their latent variables are conditionally independent, hence $p(\mathbf{z}_i|y_i) = \int_\psi p(\psi_i)p(\mathbf{z}_i|y_i; \psi_i) = \int_\psi p(\psi_i) \prod_j p(z_i^j|y_i; \psi_i) = s(\mathbf{z}_i)$, a function of the latent variables that *non-parametrically* approximates the joint distribution of latent variables of semantically related data. (Note that *un*semantically related data are independent and the joint distribution over their latent variables is a product of marginals).

We assume a Gaussian prior $p(\psi_i) = \mathcal{N}(\psi_i; 0, \gamma^2 I)$ and class-conditionals $p(z_i^j|\psi_i) = \mathcal{N}(z_i^j; \psi_i, \sigma^2)$ (for fixed variance $\sigma^2$).

$$
\begin{aligned}
p(\mathbf{z}_i|y_i) &= \int_{\psi_i} p(\mathbf{z}_i|\psi_i)p(\psi_i) = \int_{\psi_i} p(\psi_i) \prod_j p(z_i^j|\psi_i) \\
&\propto \int_{\psi_i} \exp\{-\tfrac{1}{2\gamma^2}\psi_i^2\} \prod_j \exp\{-\tfrac{1}{2\sigma^2}(z_i^j - \psi_i)^2\} \\
&= \int_{\psi_i} \exp\{-\tfrac{1}{2\sigma^2}(\tfrac{\sigma^2}{\gamma^2}\psi_i^2 + \sum_j (z_i^j - \psi_i)^2)\} \\
&= \int_{\psi_i} \exp\{-\tfrac{1}{2\sigma^2}((\sum_j z_i^{j2}) - 2(\sum_j z_i^j)\psi_i + (\tfrac{\sigma^2}{\gamma^2} + J)\psi_i^2)\} \\
&= \int_{\psi_i} \exp\{-\tfrac{\sigma^2/\gamma^2+J}{2\sigma^2}(\psi_i - \tfrac{1}{(\sigma^2/\gamma^2+J)}\sum_j z_i^j)^2\} + \exp\{-\tfrac{1}{2\sigma^2}(\sum_j z_i^{j2} - \tfrac{1}{\sigma^2/\gamma^2+J}(\sum_j z_i^j)^2)\} \quad (*) \\
&\propto \exp\{-\tfrac{1}{2\sigma^2}(\sum_j z_i^{j2} - \tfrac{1}{\sigma^2/\gamma^2+J}(\sum_j z_i^{j2} + \sum_{j\neq k} z_i^j z_i^k))\} \\
&= \exp\{-\tfrac{1}{2\sigma^2}((1 - \tfrac{1}{\sigma^2/\gamma^2+J})\sum_j z_i^{j2} + \tfrac{1}{\sigma^2/\gamma^2+J}\sum_{j\neq k} z_i^j z_i^k))\} \\
&\propto \exp\{-\tfrac{1}{2\sigma^2(\sigma^2/\gamma^2+J)}\sum_{j\neq k} z_i^j z_i^k)\} \quad\quad\quad (\text{if } \|z\|_2 = 1)
\end{aligned}
$$

The result can be rearranged into a Gaussian form (a well known result when all distributions are Gaussian), but the last line also shows that, under the common practice of setting embeddings to unit length ($\|z\|_2 = 1$), $s(\cdot)$ can be calculated directly from dot products, or cosine similarities (up to a proportionality constant, which does not affect optimisation).

If we instead assume a uniform prior, we can take the limit of the line marked (*) as $\gamma \to \infty$:

$$
\begin{aligned}
&\exp\{-\tfrac{1}{2\sigma^2}((\sum_j z_i^{j2}) - \tfrac{1}{\sigma^2/\gamma^2+J}(\sum_j z_i^j)^2)\} \\
&\to \exp\{-\tfrac{1}{2\sigma^2}((\sum_j z_i^{j2}) - \tfrac{1}{J}(\sum_j z_i^j)^2)\} \\
&= \exp\{-\tfrac{1}{2\sigma^2}((\sum_j z_i^{j2}) - J\bar{z}_i^2)\} \\
&= \exp\{-\tfrac{1}{2\sigma^2}((\sum_j z_i^{j2}) - 2J\bar{z}_i^2 + J\bar{z}_i^2)\} \\
&= \exp\{-\tfrac{1}{2\sigma^2}((\sum_j z_i^{j2}) - 2\bar{z}_i(\sum_j z_i^j) + \sum_j \bar{z}_i^2)\} \\
&= \exp\{-\tfrac{1}{2\sigma^2}\sum_j (z_i^{j2} - 2z_i^j\bar{z}_i + \bar{z}_i^2)\} \\
&= \exp\{-\tfrac{1}{2\sigma^2}\sum_j (z_i^j - \bar{z}_i)^2\}
\end{aligned}
$$

$$(8)$$

## A.5 Relationship between InfoNCE Representations and PMI

For data sampled $x \sim p(x)$ and augmentations $x' \sim p_\tau(x'|x)$ sampled under a synthetic augmentation strategy, Oord et al. (2018) show that the InfoNCE objective for a sample $x$ is optimised if their respective representations $z, z'$ satisfy

$$
\exp\{sim(z, z')\} = c \, \tfrac{p(x,x')}{p(x)p(x')}, \tag{9}
$$

where $sim(\cdot, \cdot)$ is the similarity function (e.g. dot product), and $c$ is a proportionality constant, specific to $x$. Since $c$ may differ arbitrarily with $x$ it can be considered an arbitrary function of $x$, but for simplicity we consider a particular $x$ and fixed $c$. Further, $c > 0$ is strictly positive since it is a ratio between positive (exponential) and non-negative (probability ratio) terms. Accordingly, representations satisfy

$$sim(z, z') = \text{PMI}(x, x') + c', \tag{10}$$

where $c' = \log c \in \mathbb{R}$ and $\text{PMI}(x, x')$ is the *pointwise mutual information* between samples $x$ and $x'$. Pointwise mutual information (PMI) is an information theoretic term that reflects the probability of events occurring jointly versus independently. For an arbitrary sample and augmentation this is given by:

$$\text{PMI}(x, x') \doteq \log \frac{p(x, x')}{p(x)p(x')} = \log \frac{p_\tau(x'|x)}{p(x')}. \tag{11}$$

We note that $p_\tau(x'|x) = 0$ if $x$ can *not* be augmented to produce $x'$; and that, in a continuous domain, such as images, two augmentations are identical with probability zero. Thus augmentations of different samples are expected to not overlap and the marginal is given by $p(x') = \int_x p_\tau(x'|x)p(x) = p_\tau(x'|x^*)p(x^*)$, where $x^*$ is the sample augmented to give $x'$. Thus

$$\frac{p_\tau(x'|x)}{p(x')} = \frac{p_\tau(x'|x)}{p_\tau(x'|x^*)p(x^*)} = \begin{cases} 1/p(x^*) & \dots \quad \text{if } x^* = x \text{ (i.e. } x' \text{ is an augmentation of } x) \\ 0 & \dots \quad \text{otherwise;} \end{cases} \tag{12}$$

and $\text{PMI}(x, x') = -\log p(x) \geq k > 0$ or $\text{PMI}(x, x') = -\infty$, respectively. Here $k = -\log \arg\max_x p(x)$ is a finite value based on the most likely sample. For typical datasets, this can be approximated empirically by $\frac{1}{N}$ where $N$ is the size of the original dataset (since that is how often the algorithm observes each sample), hence $k = \log N$, often of the order $5-10$ (depending on the dataset).

If the main objective were to accurately approximate PMI (subject to a constant $c'$) in Eq. 10, e.g. to approximate *mutual information*, or if representation learning *depended* on it, then, at the very least, the domain of $sim(\cdot, \cdot)$ must span its range of values, seen above as from $-\infty$ for negative samples to a small positive value (e.g. 5-10) for positive samples. Despite this, the popular *bounded* cosine similarity function $(cossim(z, z') = \frac{z^T z}{||z||_2 ||z'||_2} \in [-1, 1])$ is found to outperform the *unbounded* dot product, even though the cosine similarity function necessarily cannot span the range required to reflect true PMI values, while the dot product can. This strongly suggests that representation learning does not require representations to specifically learn PMI, or for the overall loss function to approximate mutual information.

Instead, with the cosine similarity *constraint*, the InfoNCE objective is *as optimised as possible* if representations of a data sample and its augmentations are fully aligned $(cossim(z, z') = 1)$ and representations of dissimilar data are maximally misaligned $cossim(z, z') = -1$, since these minimise the error from the true PMI values for positive and negative samples (described above). Constraints, such as the dimensionality of the representation space vs the number of samples, may prevent these revised theoretical optima being fully achieved, but the loss function is optimised by clustering representations of a sample and its augmentations and spreading apart those clusters. Note that this is the same geometric structure as induced under softmax cross-entropy loss (Dhuliawala et al., 2023).

We note that our theoretical justification for representations *not* capturing PMI is supported by the empirical observation that closer approximations of mutual information do not appear to improve representations (Tschannen et al., 2020). Also, more recent contrastive self-supervised methods increase the cosine similarity between semantically related data but spread apart representation the without negative sampling of InfoNCE, yet outperform the InfoNCE objective despite having no obvious relationship to PMI (Grill et al., 2020; Bardes et al., 2022).

### A.6   Information Loss due to Representation Collapse: a discussion

While it may seem appealing to lose information by way of representation collapse, e.g. to obtain representations *invariant* to nuisance factors, this is a problematic notion from the perspective of *general-purpose* representation

learning, where the downstream task is unknown or there may be many, since what is noise for one task may be of use in another. For example, "blur" is often considered noise, but a camera on an autonomous vehicle may be better to *detect* blur (e.g. from soiling) than be *invariant* to it and eventually fail when blurring becomes too severe. We note that humans can observe a scene *including* many irrelevant pieces of information, e.g. we can tell when an image is blurred or that we are looking through a window, and "disentangle" that from the rest of the image. This suggests that factors can be *preserved* and *disentangled*.

To stress the point that representation collapse is not desirable in and of itself, we note that collapsing together representations of semantically related data $\mathbf{x}_i$ would be problematic if subsets $\mathbf{x}_i$ overlap. For example, in the discrete case of *word2vec*, words are considered semantically related if they co-occur within a fixed window. Representation collapse, here, would mean that co-occurring words belonging to the same $\mathbf{x}_i$ would have *the same representation*, which is clearly undesirable.

## A.7 Experimental Details

### A.7.1 SimVAE Algorithm

---

**Algorithm 1** SimVAE

---

**Require:** data $\{\mathbf{x}_i\}_{i=1}^M$; batch size $N$; data dim $D$; latent dim $L$; augmentation set $\mathcal{T}$; number of augmentations $J$; encoder $f_{\phi}$; decoder $g_{\theta}$; variance of $\mathbf{z}|y$, $\boldsymbol{\sigma}^2$;
  **for** randomly sampled mini-batch $\{\mathbf{x}_i\}_{i=1}^N$ **do**
    **for** augmentation $t^j \sim \mathcal{T}$ **do**
      $\mathbf{x}_i^j = t^j(\mathbf{x}_i)$;                                                   # augment samples
      $\boldsymbol{\mu}_i^j, \boldsymbol{\Sigma}_i^j = f_{\phi}(\mathbf{x}_i^j)$;                            # forward pass: $\mathbf{z} \sim p_{\phi}(\mathbf{z}|\mathbf{x})$
      $\mathbf{z}_i^j \sim \mathcal{N}(\boldsymbol{\mu}_i^j, \boldsymbol{\Sigma}_i^j)$;
      $\tilde{\mathbf{x}}_i^j = g_{\theta}(\mathbf{z}_i^j)$;                                    # $\tilde{\mathbf{x}} = \mathbb{E}[\mathbf{x}|\mathbf{z}; \theta]$
    **end for**
    $\mathcal{L}_{\text{rec}}^i = \frac{1}{D} \sum_{j=1}^J ||\mathbf{x}_i^j - \tilde{\mathbf{x}}_i^j||_2^2$                       # minimize loss
    $\mathcal{L}_{\text{H}}^i = \frac{1}{2} \sum_{j=1}^J \log(|\boldsymbol{\Sigma}_i^j|)$
    $\mathcal{L}_{\text{prior}}^i = \frac{1}{2} \sum_{j=1}^J ||(\mathbf{z}_i^j - \frac{1}{J} \sum_{j=1}^J \mathbf{z}_i^j)/\boldsymbol{\sigma}||_2^2$
    $\min(\sum_{k=1}^N \mathcal{L}_{\text{rec}}^i + \mathcal{L}_{\text{H}}^i + \mathcal{L}_{\text{prior}}^i)$ w.r.t. $\phi, \theta$ by SGD;
  **end for**
  **return** $\phi, \theta$;

---

### A.7.2 Datasets

**MNIST.** The MNIST dataset (LeCun, 1998) gathers 60'000 training and 10'000 testing images representing digits from 0 to 9 in various caligraphic styles. Images were kept to their original 28x28 pixel resolution and were binarized. The 10-class digit classification task was used for evaluation.

**FashionMNIST.** The FashionMNIST dataset (Xiao et al., 2017) is a collection of 60'000 training and 10'000 test images depicting Zalando clothing items (i.e., t-shirts, trousers, pullovers, dresses, coats, sandals, shirts, sneakers, bags and ankle boots). Images were kept to their original 28x28 pixel resolution. The 10-class clothing type classification task was used for evaluation.

**CIFAR10.** The CIFAR10 dataset (Krizhevsky et al., 2009) offers a compact dataset of 60,000 (50,000 training and 10,000 testing images) small, colorful images distributed across ten categories including objects like airplanes, cats, and ships, with various lighting conditions. Images were kept to their original 32x32 pixel resolution.

**CelebA.** The CelebA dataset (Liu et al., 2015) comprises a vast collection of celebrity facial images. It encompasses a diverse set of 183'000 high-resolution images (i.e., 163'000 training and 20'000 test images),

each depicting a distinct individual. The dataset showcases a wide range of facial attributes and poses and provides binary labels for 40 facial attributes including hair & skin colour, presence or absence of attributes such as eyeglasses and facial hair. Each image was cropped and resized to a 64x64 pixel resolution. Attributes referring to hair colour were aggregated into a 5-class attribute (i.e., bald, brown hair, blond hair, gray hair, black hair). Images with missing or ambiguous hair colour information were discarded at evaluation.

All datasets were sourced from Pytorch's dataset collection.

### A.7.3 Data augmentation strategy

Taking inspiration from SimCLR's (Chen et al., 2020a) augmentation strategy which highlights the importance of random image cropping and colour jitter on downstream performance, our augmentation strategy includes random image cropping, random image flipping and random colour jitter. The colour augmentations are only applied to the non gray-scale datasets (i.e., CIFAR10 & CelebA datasets). Due to the varying complexity of the datasets we explored, hyperparameters such as the cropping strength were adapted to each dataset to ensure that semantically meaningful features remained after augmentation. The augmentation strategy hyperparameters used for each dataset are detailed in table 2.

| Dataset | Crop | | Vertical Flip | Colour Jitter | | |
|---------|------|--|---------------|---------------|--|--|
|  | scale | ratio | prob. | b-s-c | hue | prob. |
| MNIST | 0.4 | [0.75,1.3] | 0.5 | - | - | - |
| Fashion | 0.4 | [0.75,1.3] | 0.5 | - | - | - |
| CIFAR10 | 0.6 | [0.75,1.3] | 0.5 | 0.8 | 0.2 | 0.8 |
| CelebA | 0.6 | [0.75,1.3] | 0.5 | 0.8 | 0.2 | 0.8 |

Table 2: Data augmentation strategy for each dataset: (left to right) cropping scale, cropping ratio, probability of vertical flip, brightness-saturation-contrast jitter, hue jitter, probability of colour jitter

### A.7.4 Training Implementation Details

This section contains all details regarding the architectural and optimization design choices used to train SimVAE and all baselines. Method-specific hyperparameters are also reported below.

**Network Architectures.** The encoder network architectures used for SimCLR, MoCo, VicREG, and VAE-based approaches including SimVAE for simple (i.e., MNIST, FashionMNIST ) and complex datasets (i.e., CIFAR10, CelebA) are detailed in Table 3a, Table 4a respectively. Generative models which include all VAE-based methods also require decoder networks for which the architectures are detailed in Table 3b and Table 4b. The latent dimensionality for MNIST and FashionMNIST is fixed at 10 and increased to 64 for the CelebA and CIFAR10 datasets. The encoder and decoder architecture networks are kept constant across methods including the latent dimensionality to ensure a fair comparison.

| Layer Name | Output Size | Block Parameters | Layer Name | Output Size | Block Parameters |
|------------|-------------|------------------|------------|-------------|------------------|
| fc1 | 500 | 784x500 fc, relu | fc1 | 2000 | 10x2000 fc, relu |
| fc2 | 500 | 500x500 fc, relu | fc2 | 500 | 2000x500 fc, relu |
| fc3 | 2000 | 500x2000 fc, relu | fc3 | 500 | 500x500 fc, relu |
| fc4 | 10 | 2000x10 fc | fc4 | 784 | 500x784 fc |

(a) Encoder  (b) Decoder

Table 3: Multi-layer perceptron network architectures used for MNIST & FashionMNIST training

**Optimisation & Hyper-parameter tuning.** All methods were trained using an Adam optimizer until training loss convergence. The batch size was fixed to 128. Hyper-parameter tuning was performed based

| Layer Name | Output | Block Parameters | Layer Name | Output | Block Parameters |
|---|---|---|---|---|---|
| conv1 | 32x32 | 4x4, 16, stride 1 
 batchnorm, relu 
 3x3 maxpool, stride 2 | fc | 256x4x4 | 64x4096 fc |
| conv2_x | 32x32 | 3x3, 32, stride 1 
 3x3, 32, stride 1 | conv1_x | 8x8 | 3x3, 128, stride 2 
 3x3, 128, stride 1 |
| conv3_x | 16x16 | 3x3, 64, stride 2 
 3x3, 64, stride 1 | conv2_x | 16x16 | 3x3, 64, stride 2 
 3x3, 64, stride 1 |
| conv4_x | 8x8 | 3x3, 128, stride 2 
 3x3, 128, stride 1 | conv3_x | 32x32 | 3x3, 32, stride 2 
 3x3, 32, stride 1 |
| conv5_x | 4x4 | 3x3, 256, stride 2 
 3x3, 256, stride 1 | conv4_x | 64x64 | 3x3, 16, stride 2 
 3x3, 16, stride 1 |
| fc | 64 | 4096x64 fc | conv5 | 64x64 | 5x5, 3, stride 1 |

(a) Encoder                                                   (b) Decoder

Table 4: Resnet18 network architectures used for CIFAR10 & CelebA datasets

on the downstream MLP classification accuracy across datasets. The final values of hyperparameters were selected to reach the best average downstream performance across datasets. While we observed stable performances across datasets for the VAE family of models, VicREG and MoCo, SimCLR is more sensitive, leading to difficulties when having to define a unique set of parameters across datasets. For VAEs, the learning rate was set to $8e^{-5}$, and the likelihood probability, $p(x|z)$, variance parameter was set to 0.02 for $\beta$-VAE, CR-VAE and SimVAE. CR-VAE's $\lambda$ parameter was set to 0.1. SimVAE's prior probability, $p(z|y)$, variance was set to 0.15 and the number of augmentations to 10. VicREG's parameter $\mu$ was set to 25 and the learning rate to 1e-4. SimCLR's temperature parameter, $\tau$, was set to 0.7 and learning rates were adapted for each dataset due to significant performance variations across datasets ranging from $8e^{-5}$ to $1e^{-3}$. For MoCo, the temperature parameter was fixed to 0.7, for MoCov2 to 0.2. To ensure a more fair comparison between methods, we kept the augmentation strategy identical across methods and did not add blurring for MoCov2 training.

### A.7.5 Evaluation Implementation Details

Following common practices (Chen et al., 2020a), downstream performance is assessed using a linear probe, a multi-layer perceptron probe, a $k$-nearest neighbors (kNN) algorithm, and a Gaussian mixture model (GMM). The linear probe consists of a fully connected layer whilst the mlp probe consists of two fully connected layers with a relu activation for the intermediate layer. Both probes were trained using an Adam optimizer with a learning rate of $3e^{-4}$ for 200 epochs with batch size fixed to 128. Scikit-learn's Gaussian Mixture model with a full covariance matrix and 200 initialization was fitted to the representations using the ground truth cluster number. The $k$-NN algorithm from Python's Scikit-learn library was used with k spanning from 1 to 15 neighbors. The best performance was chosen as the final performance measurement. No augmentation strategy was used at evaluation.

**Computational Resources.** Models for MNIST, FashionMNIST and CIFAR10 were trained on a RTX2080ti GPU with 12G RAM. Models for CelebA were trained on an RTX3090 GPU with 24G RAM. We observe that while the family of generative models requires more time per iteration, the loss converges faster while discriminative methods converge at a slower rate when considering the optimal set of hyperparameters. As a consequence, generative baselines and SimVAE were trained for 400 epochs while discriminative methods were trained for 600 to 800 epochs.

### A.7.6 Generation Protocol

Here we detail the image generation protocol and the quality evaluation of generated samples.

| | Acc-LP | | | | Acc-GMM | | | |
|---|---|---|---|---|---|---|---|---|
| | MNIST | Fashion | CelebA | CIFAR10 | MNIST | Fashion | CelebA | CIFAR10 |
| Random | 39.7 ± 2.4 | 51.2 ± 0.6 | 64.4 ± 0.9 | 15.7 ± 0.9 | 42.2 ± 1.2 | 48.6 ± 0.2 | 59.2 ± 0.3 | 13.1 ± 0.6 |
| SimCLR | **96.8** ± 0.1 | **73.0** ± 0.3 | 94.2 ± 0.2 | 65.4 ± 0.1 | **83.7** ± 0.6 | 53.6 ± 0.3 | **71.6** ± 0.6 | 28.2 ± 0.2 |
| MoCo | 88.6 ± 1.7 | 65.0 ± 1.3 | - | 53.3 ± 1.3 | 70.5 ± 4.0 | 56.6 ± 1.1 | - | **52.4** ± 0.3 |
| VicREG | 96.7 ± 0.0 | 71.7 ± 0.1 | **94.3** ± 0.3 | **68.2** ± 0.0 | 79.8 ± 0.6 | **60.2** ± 1.1 | 53.9 ± 0.2 | 35.0 ± 2.8 |
| VAE | 97.2 ± 0.2 | 79.0 ± 0.5 | 81.5 ± 1.0 | 24.7 ± 0.4 | 96.3 ± 0.4 | 57.9 ± 0.8 | 58.8 ± 0.2 | 23.4 ± 0.0 |
| $\beta$-VAE | 97.8 ± 0.0 | 79.6 ± 0.0 | 81.9 ± 0.2 | 26.9 ± 0.0 | 96.2 ± 0.2 | 68.0 ± 0.3 | 59.5 ± 0.6 | 31.2 ± 0.1 |
| CR-VAE | 97.5 ± 0.0 | **79.7** ± 0.0 | 81.6 ± 0.3 | 26.8 ± 0.0 | **96.9** ± 0.0 | 63.4 ± 0.4 | 58.9 ± 0.4 | 30.3 ± 0.0 |
| SimVAE | **98.0** ± 0.0 | **80.0** ± 0.0 | **87.1** ± 0.3 | **40.1** ± 0.0 | 96.6 ± 0.0 | **71.1** ± 0.0 | 58.4 ± 0.6 | **39.3** ± 0.0 |

Table 5: Top-1% self-supervised Acc (↑) for MNIST, FashionMNIST, CIFAR10, and CelebA (gender classification) using a linear probe (LP) and Gaussian Mixture Model (GMM) classification methods; We report mean and standard errors over three runs; Bold indicate best scores in each method class: generative (teal), discriminative methods (red).

**Ad-hoc decoder training** VAE-based approaches, including SimVAE, are fundamentally generative methods aimed at approximating the logarithm of the marginal likelihood distribution, denoted as $\log p(x)$. In contrast, most traditional self-supervised methods adopt a discriminative framework without a primary focus on accurately modeling $p(x)$. However, for the purpose of comparing representations, and assessing the spectrum of features present in $z$, we intend to train a decoder model for SimCLR & VicREG models. This decoder model is designed to reconstruct images from the fixed representations initially trained with these approaches. To achieve this goal, we train decoder networks using the parameter configurations specified in Tables 3b and 4b, utilizing the mean squared reconstruction error as the loss function. The encoder parameters remain constant, while we update the decoder parameters using an Adam optimizer with a learning rate of $1e^{-4}$ until a minimal validation loss is achieved (i.e. $\sim$ 10-80 epochs).

**Conditional Image Generation.** To allow for a fair comparison, all images across all methods are generated by sampling $z$ from a multivariate Gaussian distribution fitted to the training samples' representations. More precisely, each Gaussian distribution is fitted to $z$ conditioned on a label $y$. Scikit-Learn Python library Gaussian Mixture model function (with full covariance matrix) is used.

## A.8 Additional Results & Ablations

**Content retrieval with linear & gaussian mixture model prediction heads.** ?? reports the top-1% self-supervised classification accuracy using a linear prediction head and a gaussian mixture model. From ??, we draw similar conclusion as with Table 1: SimVAE significantly bridges the gap between discriminative and generative self-supervised learning methods when considering a supervised linear predictor and fully unsupervised methods for downstream prediction. Table 6 report the normalized mutual information (NMI) and adjusted rank index (ARI) for the fitting of the GMM prediction head.

**Content & Style retrieval for all CelebA attributes.** Figure 6 reports average classification accuracy using a MLP probe (over 3 runs) for the prediction of 20 CelebA facial attributes for SimVAE, generative and discriminative baselines.

**Augmentation protocol strength ablation.** Figure 7 reports the downstream classification accuracy across methods for various augmentations strategies. More precisely, we progressively increase the cropping scale and color jitter amplitude. Unsurprisingly (Chen et al., 2020a), discriminative methods exhibit high sensitivity to the augmentation strategy with stronger disruption leading to improved content prediction. The opposite trend is observed with vanilla generative methods where reduced variability amongst the data leads to increased downstream performance. Interestingly, SimVAE is robust to augmentation protocol and performs comparably across settings.

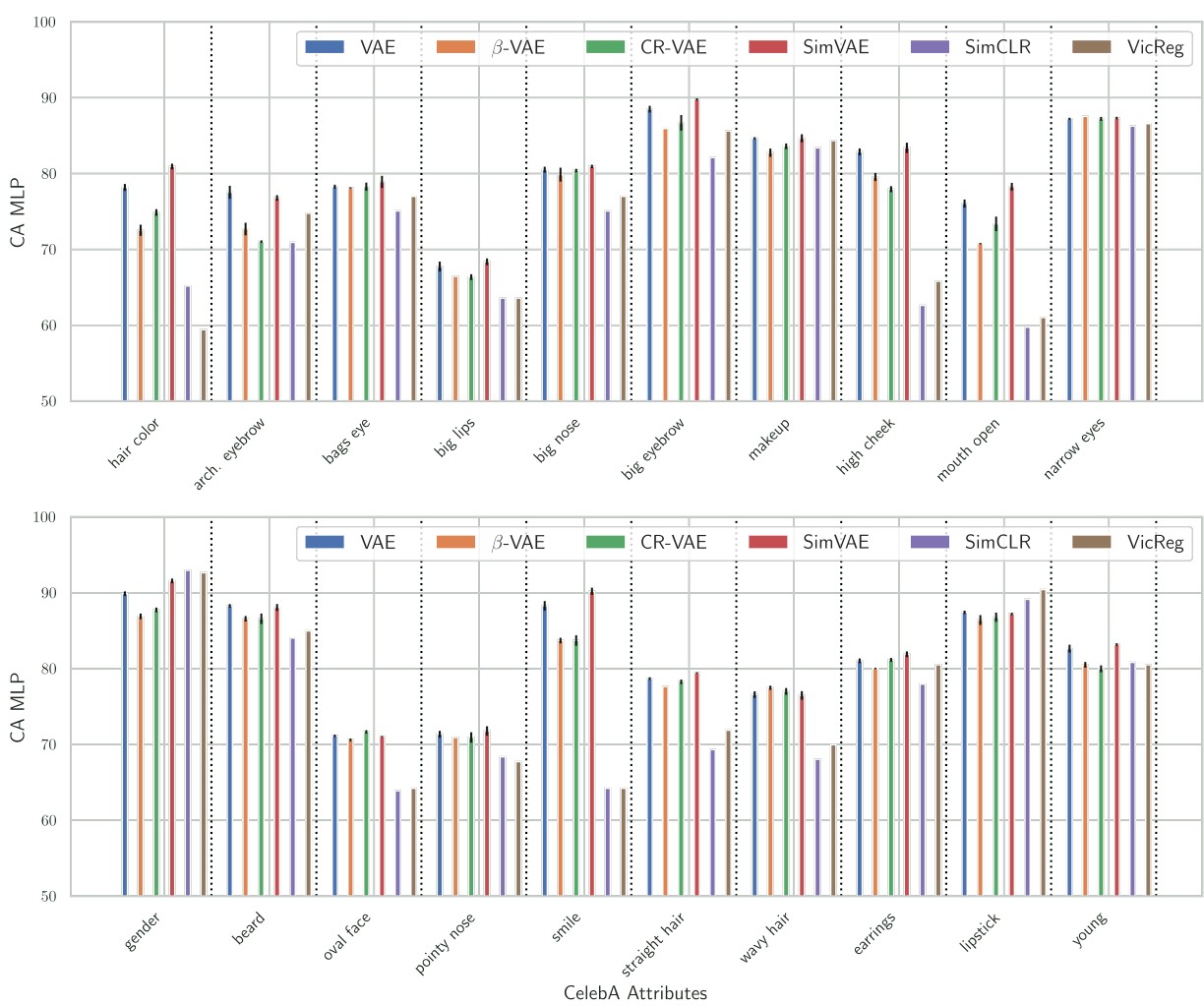

Figure 6: CelebA 20 facial attributes prediction using a MP. Average scores and standard errors are reported across 3 random seeds.

| Dataset | | VAE | $\beta$-VAE | CR-VAE | SimVAE | MoCo | VicREG | SimCLR |
|---|---|---|---|---|---|---|---|---|
| **MNIST** | ARI | 89.0 ± 1.0 | 93.3 ± 0.3 | **94.0** ± **0.0** | 93.1 ± 0.0 | 58.3 ± 3.8 | 72.0 ± 0.7 | 77.4 ± 0.2 |
| | NMI | 94.9 ± 0.4 | 96.7 ± 0.2 | **96.9** ± **0.0** | 96.6 ± 0.0 | 71.4 ± 2.5 | 86.8 ± 0.4 | 89.6 ± 0.1 |
| **Fashion** | ARI | 44.3 ± 0.9 | 53.3 ± 0.4 | 47.6 ± 0.4 | **56.8** ± **0.0** | 30.9 ± 0.5 | 41.2 ± 0.5 | 33.2 ± 0.3 |
| | NMI | 69.1 ± 0.6 | 75.6 ± 0.1 | 72.6 ± 0.1 | **77.1** ± **0.0** | 50.4 ± 0.6 | 66.9 ± 0.3 | 62.1 ± 0.2 |
| **CelebA** | ARI | 5.7 ± 0.2 | 6.2 ± 0.7 | 6.6 ± 0.9 | 2.6 ± 0.7 | − | **18.7** ± **0.8** | 0.0 ± 0.1 |
| | NMI | 3.9 ± 0.2 | 4.7 ± 0.9 | 5.0 ± 0.7 | 2.9 ± 0.7 | − | **24.3** ± **0.3** | 0.0 ± 0.0 |
| **CIFAR10** | ARI | 0.6 ± 0.0 | 2.9 ± 0.1 | 2.0 ± 0.0 | 12.2 ± 0.1 | 27.2 ± 1.0 | 25.7 ± 0.2 | **52.2** ± **0.1** |
| | NMI | 31.7 ± 0.0 | 33.5 ± 0.1 | 32.4 ± 0.0 | 42.8 ± 0.1 | 16.5 ± 0.4 | **55.3** ± **0.1** | 21.7 ± 0.1 |

Table 6: Normalized mutual information (NMI) and Adjusted Rank Index (ARI) for all methods and datasets; Average scores and standard errors are computed across three runs

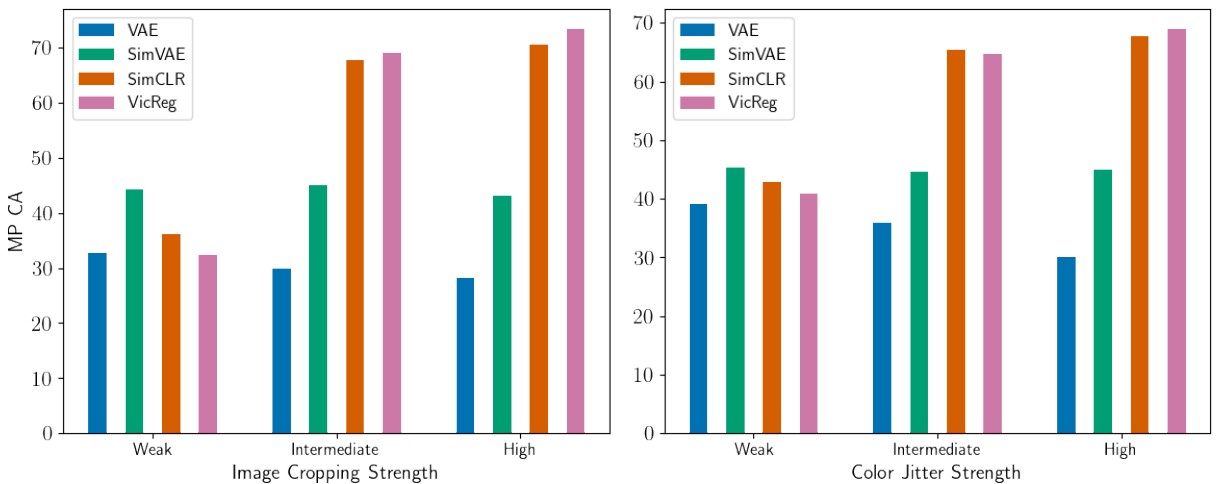

Figure 7: Ablation experiment across the augmentation strength for (*left*) image cropping and (*right*) color jitter strength considered during training of the SimVAE and baseline models using the CIFAR dataset.

**# of augmentation ablation.** Figure 8 reports the downstream classification accuracy for increasing numbers of augmentations considered simultaneously during the training of SimVAE for MNIST and CIFAR10 datasets. On average, a larger number of augmentations result in a performance increase. Further exploration is needed to understand how larger sets of augmentations can be effectively leveraged potentially by allowing for batch size increase. From Figure 8, we fix our number of augmentations to 10 across datasets.

**Likelihood $p(x|z)$ variance ablation.** We explore the impact of the likelihood, $p(x|z)$, variance, $\sigma^2$, across each pixel dimension on the downstream performance using the MNIST and CIFAR10 datasets. Figure 9 highlights how the predictive performance is inversely correlated with the $\sigma^2$ on the variance range considered for the CIFAR10 dataset. A similar ablation was performed on all VAE-based models and led to a similar conclusion. We therefore fixed $\sigma^2$ to 0.02 for $\beta$-VAE, CR-VAE and SimVAE across datasets.

**Generated Images.** Figure 10 report examples of randomly generated images for each digit class and clothing item using the SimVAE trained on MNIST FashionMNIST, CIFAR10 and CelebA respectively.

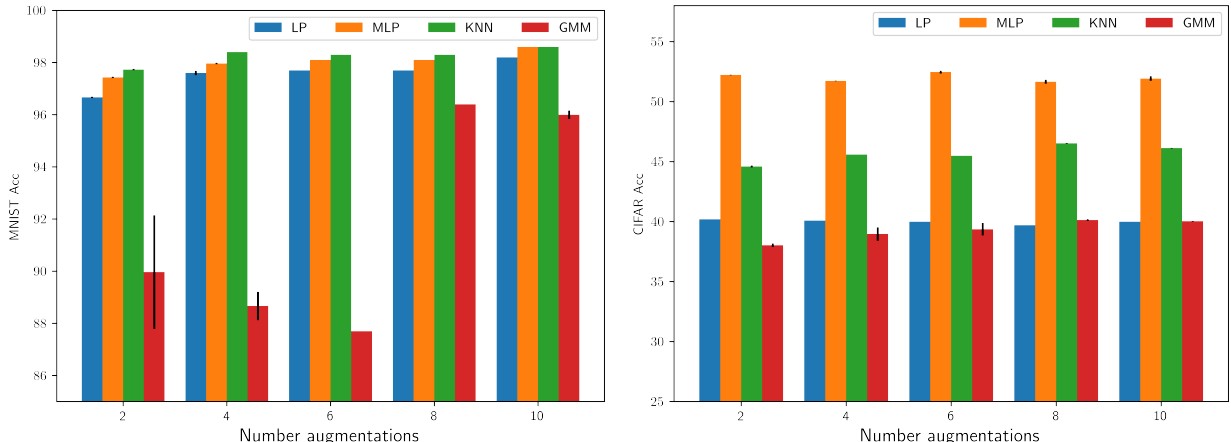

Figure 8: Ablation experiment across the number of augmentations considered during training of the SimVAE model using the MNIST (left) and CIFAR10 (right) datasets. Two, four, six, eight, and 10 augmentations were considered. The average and standard deviation of the downstream classification accuracy using Linear, MLP probes, and a KNN & GMM estimators are reported across three seeds. Batch size of 128 for all reported methods and number of augmentations. Means and standard errors are reported for three runs.

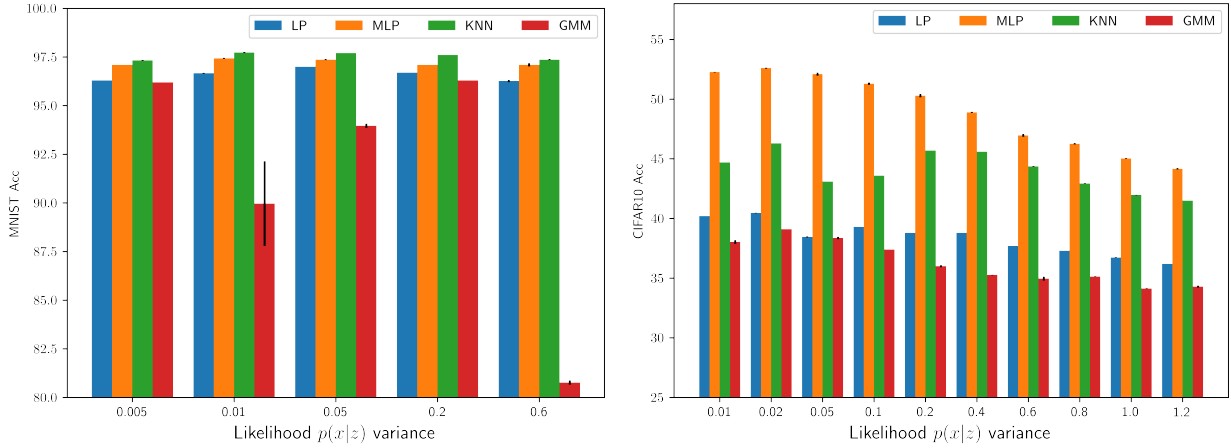

Figure 9: Ablation experiment across the likelihood $p(x|z)$ variance considered during training of the SimVAE model using the MNIST (left) and CIFAR10 (right) datasets. The average and standard deviation of the downstream classification accuracy using Linear, MLP probes and a KNN & GMM estimators are reported across three seeds. Means and standard errors are reported for three runs.

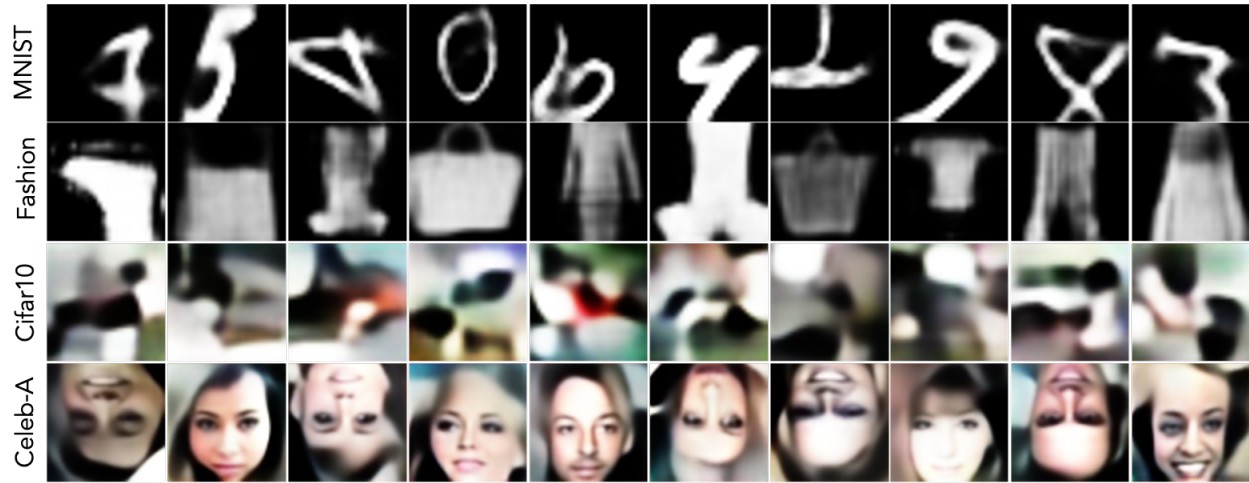

Figure 10: Samples generated from SimVAE model using MNIST, FashionMNIST, Cifar10 and CelebA training datasets

**Generative Quality.** Table 7 reports the FID scores and reconstruction error for all generative baselines and SimVAE.

|  | MNIST | Fashion | CelebA | CIFAR10 |
|---|---|---|---|---|
| **MSE (↓)** | | | | |
| VAE | $0.029 \pm 0.0$ | $0.012 \pm 0.0$ | $0.016 \pm 0.0$ | $0.008 \pm 0.0$ |
| $\beta$-VAE | $0.029 \pm 0.0$ | $\mathbf{0.008} \pm 0.0$ | $0.005 \pm 0.0$ | $0.004 \pm 0.0$ |
| CR-VAE | $0.030 \pm 0.0$ | $\mathbf{0.008} \pm 0.0$ | $0.005 \pm 0.0$ | $0.004 \pm 0.0$ |
| SimVAE | $\mathbf{0.026} \pm 0.0$ | $0.009 \pm 0.0$ | $\mathbf{0.004} \pm 0.0$ | $\mathbf{0.003} \pm 0.0$ |
| **FID (↓)** | | | | |
| VAE | $\mathbf{150.1} \pm 0.2$ | $99.4 \pm 0.6$ | $162.9 \pm 2.8$ | $365.4 \pm 3.3$ |
| $\beta$-VAE | $155.3 \pm 0.5$ | $99.9 \pm 0.7$ | $163.8 \pm 2.3$ | $376.7 \pm 1.7$ |
| CR-VAE | $153.0 \pm 0.9$ | $98.7 \pm 0.0$ | $\mathbf{159.3} \pm 5.4$ | $374.4 \pm 0.4$ |
| SimVAE | $152.7 \pm 0.3$ | $\mathbf{96.1} \pm 1.0$ | $\mathbf{157.8} \pm 2.3$ | $\mathbf{349.9} \pm 2.1$ |

Table 7: Generation quality evaluated by: mean squared reconstruction error (MSE), Fréchet inception distance (FID). Mean and standard errors are reported across three runs.

