# OpenReview forum: "A Probabilistic Model behind Self- Supervised Learning"
_TMLR — Accepted by TMLR_

### Review · Reviewer_3vMd · 2024-06-30

**Summary Of Contributions:**

This paper introduces a generative latent variable model for self-supervised learning, demonstrating that various types of discriminative self-supervised learning (SSL) methods generate a comparable distribution over representations. This offers a unifying theoretical framework for these approaches. Additionally, the proposed model elucidates the relationship with mutual information and the application of a "projection head." Experimental results indicate that SimVAE outperforms other VAE-based models.

**Audience:**

Yes

**Broader Impact Concerns:**

No societal concerns.

**Claims And Evidence:**

Yes

**Requested Changes:**

See weakness.

**Strengths And Weaknesses:**

***Strengths***:
1. This paper is well-structured and well-written.
2. Theoretical analysis makes this study more solid.
3. SimVAE shows better performance than the considered VAE-based methods.

***Weakness***:
1. While SimVAE shows theoretical promise, it is currently not as competitive as non-VAE-based baselines such as MoCo, and maybe other stronger methods, such as MoCoV2 [a], and MAE [b]. Given the current experimental results, it may be premature to fully assert the potential of VAE-based methods.
2. The current experiments are still relatively small scale. It would be interesting to know the performance on more complex tasks (a larger number of classes and/or the distribution is harder to model), such as ImageNet. Does the model perform consistently and stably better than other methods?
3. It is unclear why MoCo is not analysed in Figure 5. The analysis on "Content" vs. "Style" is interesting. However, more experiments (more datasets) is needed to validate the generalisability of conclusions.
3. Referring to "uncertainty estimation" is confusing, as there is already a research area dedicated to aligning model confidence with model performance.

4. Typos:

   1. In Abstract: supevised $\rightarrow$ supervised

   2. abd $\rightarrow$ and

[a] Improved baselines with momentum contrastive learning, arXiv 2020.

[b] Masked Autoencoders Are Scalable Vision Learners, CVPR 2022.

---

> ### Author Response · Authors · 2024-07-23
> **Response to review - Part 1**
>
> Thank you for your review and acknowledging that our work is **well-written and structured**, **sound** and that the proposed model is **effective**. Below, we address your comments and implement your suggestions which help improve our work.
>
> - _“Premature to fully assert the potential of VAE-based methods as stronger baselines like MoCo v2 and MAE are missing”_:
>     - Thank you for your suggestions. Regarding the stronger baselines:
>         - MoCo v2 [2]: MoCo v2 modifies various aspects of  MoCo, e.g. the data augmentation strategy, learning rate schedule, temperature parameter and uses an MLP projection head rather than linear. For fair comparison, our experimental setup uses the same augmentation strategy for all SSL methods. As such, following your suggestion we add MoCo v2 to our evaluation with the same augmentation strategy as used for other methods. The results are included in Table 1 and Figure 5 of the updated manuscript and the general conclusions drawn from them are unchanged. The average performance over all attributes for MoCo v2 is similar to SimVAE, but Fig 5 (center) shows a clear difference when we consider specific  attributes relating to style vs content.
>         - MAE [1]: masked autoencoders use transformer architectures, which we do not consider in this work as we focus on understanding the training objectives of SSL methods vs our proposed ELBO, and keep the model architecture constant for fair comparison. (Note that attempts to achieve MAE performance with convolutional networks require significant other changes to the architecture of loss function Since one of our conclusions is that generative representation learning offers potential benefits but is more challenging for a given architecture, we agree that considering more complex model architectures, e.g. transformers, is a natural future extension of our current, more theoretically focused work.
>     - Regarding "potential of VAE-based methods", we believe that our analysis and results suggest the *potential* of generative representation learning. While further research is required for such potential to be realised, this theoretically principled approach achieves comparable or improved results to discriminative methods on simpler data sets (Table 1), but avoids the information loss that discriminative representation learning can suffer from.
> - _"experiments are still relatively small scale"_:
>     -  Thank you, we agree that it is important for representation learning methods to scale, and for our choice of datasets and limitations of our work to be made clear.
>     - Firstly, the primary aim of our work is to propose a theroretical basis for families of SSL methods to give insight into how they work. Our analysis identifies that certain information ("style") may be lost by self-supervised methods, which our experiments aim to demonstrate. Our analysis also suggets that learning representations generatively by maximising the ELBO (rather than a discriminative proxy to it) may overcome this information loss, which we demonstrate on the same data sets.
>     - However, learning representations for complex data distributions under a generative regime remains a challenge  relative to discriminative approaches. In particular, a generative model must capture sufficient information to fully regenerate an image, not only enough to discriminate samples; and must model complex distributions $p(x|z)$ in the (typically) high-dimensional data space. Recent work [1] quantifies this by showing that features that succesfully predict labels may capture only a little of the variance in the data, which a generative model must capture in full. This demonstrates that discriminative models have an "easier job" relative to generative approaches and justifies why the latter are less competitive on more complex datasets, e.g. ImageNet, given the same model architecture. Promisingly, [1] suggests ways in which these issues might be alleviated in the future and generative modelling has recently seen significant advances, e.g. diffusion models, suggesting that generative representation learning may greatly improve with future research.
>     - Thus, our choice of data sets is chosen (i) to demonstrate the key findings of our analysis, in particular that discriminative methods lose information and a generative approach can learn useful representations, while (ii) being capable of being modelled by VAE-based models (the datasets we use, MNIST, FashionMNIST and CelebA, are reference benchmarks for VAE-based methods [e.g. 2, 3]).
>     - In summary, by providing a theoretical rationale for self-supervised representation learning, which reveals pitfalls in current discriminative methods and suggests an alternative principled generative approach, we hope to encourage future research in the generative direction that is able to overcome the orthogonal challenges a generative approach brings.
>     - We have updated Section 8 to convey this more clearly.

---

> ### Author Response · Authors · 2024-07-23
> **Response to review - Part 2**
>
> - _"MoCo is not analysed in Figure 5"_:
>     - Thank you for raising this omission, we have updated figure 5 to include the MoCo baseline. The results show that MoCo achieves improved style recovery compared to other discriminative SSL, but do not change the general conclusions in results section 7: SimVAE significantly better than predictive SSL on style attributes such as hair color.
> - _"experiments on style recovery are limited"_:
>     - We agree that further empirical evidence of the performance gap between discriminative and generative SSL at style prediction would increase confidence in this finding. However, we are unaware of other datasets with multiple semantic annotations to demonstrate this, comparable to CelebA. For example, we considered ChestMNIST [5] (chest X-rays with annotations of 14 pulmonary diseases), but preliminary results (below) show that both discriminative and generative SSL methods perform very poorly and no conclusions regarding style vs content can be drawn. Importantly, we note that the difference in ability to identify style information from discriminatively and generatively learned representations is predicted from our theory and it is notable that the one dataset we demonstrate this on shows a clear difference.
>     - We also include qualitative comparisons of discriminatively and generatively learned representations by visualising  reconstructions on four datasets (Figs 3, 4) which suggest similar conclusions.
>
> |     Disease classification               | Atelectasis | Cardiomegaly | Effusion | Infiltration |
> |--------------------|-------------|--------------|----------|--------------|
> | *% positive samples* | $89.80\%$     | $97.50\%$       | $88.20\%$  | $82.20\%$      |
> | VAE                | $89.21\%$    | $97.40\%$     | $87.80\%$ | $82.50\%$      |
> | CR-VAE             | $89.21\%$    | $97.46\%$     | $87.80\%$ | $82.40\%$     |
> | SimCLR             | $89.22\%$    | $97.40\%$     | $87.84\%$ | $82.44\%$     |
> | VicReg             | $89.30\%$     | $97.40\%$      | $87.87\%$ | $82.42\%$     |
>
> Table 1. Self-supervised downstream binary classification accuracy for 4 pulmonary diseases from the ChestMNIST dataset, for predictive and generative SSL baselines. Performance across methods is broadly equivalent to that of a classifier that always predicts the positive class. Similar behaviour is observed for the 10 other ChestMNIST tasks.
>
> - _"referring to uncertainty estimation is confusing"_
>     - Apologies for any confusion. We refer to "uncertainty estimation" as a potential benefit of generative representation learning since representations are generated with a measure of their uncertainty. Specifically, the posterior $q(z|x)$ under the generative model's ELBO can be viewed as a measure of uncertainty of a representation (typically chosen as the mean of the posterior). This has practical benefit, for example less confidence should be attributed to downstream predictions based on a representation with a high variance posterior $q(z|x)$. Alternatively, multiple representations could be sampled from the posterior to give a Bayesian estimate of a downstream property.
> - Typos: thank you, these have been fixed.
>
> [1] He, Kaiming, et al. "Masked autoencoders are scalable vision learners." Proceedings of the IEEE/CVF conference on computer vision and pattern recognition. 2022.
>
> [2] Chen, Xinlei, et al. "Improved baselines with momentum contrastive learning." arXiv preprint arXiv:2003.04297 (2020).
>
> [3] Li, Siyuan, et al. "Architecture-Agnostic Masked Image Modeling-From ViT back to CNN." Proceedings of Machine Learning Research 202 (2023): 19460-19470.
>
> [4] Gao, Peng, et al. "Convmae: Masked convolution meets masked autoencoders." arXiv preprint arXiv:2205.03892 (2022).

---

### Review · Reviewer_43pU · 2024-07-03

**Summary Of Contributions:**

This paper proposes a latent variable model as a theoretical framework to understand self-supervised learning (SSL) methods, particularly contrastive learning objectives. The model is learned by maximizing the Evidence Lower Bound (ELBO), which the authors use to connect their framework with existing SSL methods. They introduce SimVAE with a mixture of Gaussian prior to bridge the gap between generative and predictive SSL.

**Audience:**

Yes

**Broader Impact Concerns:**

I don't see any broader impact concerns.

**Claims And Evidence:**

Yes

**Requested Changes:**

1. In SimVAE with Gaussian mixture prior, please clarify how the number of clusters is defined. Is a uniform prior assumed for p(y)?

2. While bridging the gap between generative and predictive SSL is valuable, the paper could benefit from more discussion on fundamental insights to improve SSL beyond simply adding a reconstruction term. Adding reconstruction term will increase the model parameters and also perhaps instable training.

**Strengths And Weaknesses:**

Strengths:

1. The paper provides a comprehensive review of related work and uses the ELBO to connect predictive self-supervised learning with generative learning.

2. The paper is well-structured and easy to follow, with clear and straightforward derivations.

3. The empirical results show SimVAE is competitive with or outperforms several existing benchmarks, particularly on simpler datasets and tasks requiring style information.

Weaknesses:

1. The empirical evaluation is limited to relatively small and simple datasets.

2. While the rationale behind the projection head is interesting, it lacks substantial evidence to support the claim that it brings representations closer to the SSL prior.

---

> ### Author Response · Authors · 2024-07-23
> **Answer to Reviewer43pU -Part 1**
>
> Thank you for your review and for acknowledging that our work is **well-written**, **principled**, that **derivations are clear**, and that empirical results demonstrate the impact on style information. We address your comments below and have implemented changes in the manuscript following your suggestions, which have improved our work:
>
> - _“relatively small and simple datasets”_:
>     -  Thank you, we agree that it is important for representation learning methods to scale, and for our choice of datasets and limitations of our work to be made clear.
>     - Firstly, the primary aim of our work is to propose a theroretical basis for families of SSL methods to give insight into how they work. Our analysis identifies that certain information ("style") may be lost by self-supervised methods, which our experiments aim to demonstrate. Our analysis also suggests that learning representations generatively by maximising the ELBO (rather than a discriminative proxy to it) may overcome this information loss, which we demonstrate on the same data sets.
>     - However, learning representations for complex data distributions under a generative regime remains a challenge relative to discriminative approaches. In particular, a generative model must capture sufficient information to fully regenerate an image, not only enough to discriminate samples; and must model complex distributions $p(x|z)$ in the (typically) high-dimensional data space. Recent work [1] quantifies this by showing that features that successfully predict labels may capture only a little of the variance in the data, which a generative model must capture in full. This demonstrates that discriminative models have an "easier job" relative to generative approaches and justifies why the latter are less competitive on more complex datasets, e.g. ImageNet, given the same model architecture. Promisingly, [1] suggests ways in which these issues might be alleviated in the future and generative modelling has recently seen significant advances, e.g. diffusion models, suggesting that generative representation learning may greatly improve with future research.
>     - Thus, our choice of data sets is chosen (i) to demonstrate the key findings of our analysis, in particular that discriminative methods lose information and a generative approach can learn useful representations, while (ii) being capable of being modelled by VAE-based models (the datasets we use, MNIST, FashionMNIST and CelebA, are reference benchmarks for VAE-based methods [e.g. 2, 3]).
>     - In summary, by providing a theoretical rationale for self-supervised representation learning, which reveals pitfalls in currrent disciminative methods and suggests an alternative principled generative approach, we hope to encourage future research in the generative direction that is able to overcome the othogonal challenges a generative approach brings.
>     - We have updated Section 8 to convey this more clearly.
>
> [1] Balestriero, Randall, and Yann LeCun. "How Learning by Reconstruction Produces Uninformative Features For Perception." Forty-first International Conference on Machine Learning.
>
> [2] Manduchi, Laura, et al. "Tree variational autoencoders." *Advances in Neural Information Processing Systems* 36 (2023): 54952-54986.
>
> [3] Sinha, Samarth, and Adji Bousso Dieng. "Consistency regularization for variational auto-encoders." *Advances in Neural Information Processing Systems* 34 (2021): 12943-12954.

---

> > ### Author Response · Authors · 2024-07-23
> > **Answer to Reviewer 43pU**
> >
> > - _“rationale behind the projection head is interesting but lacks substantial evidence”_:
> >     - Previous works have shown empirically: (i) that using a projection head improves performance [e.g. 1]; and (ii) that inputs to the projection head have higher "intra-class feature variation" relative to projection head outputs, or when no projection head is used [1,2,3]. In particular:
> >         - [1] introduced the projection head and conjectured that it may improve performance due to a "loss of information induced by contrastive loss", which "removes information that may be useful for the downstream task". Outputs of the projection head were shown to be approximately low-rank (Fig B.3).
> >         - [2] illustrate (Fig 4) that representations have a higher variance when taken as the input to the projection head, compared to not using a projection head.
> >         - [3] also illustrate that representations have higher intra-class class variance when a projection head is used (Fig 1) and show quantitatively (from covariance matrix eigenvalues) that inputs of a projection head have higher variance than its output, or when no projection head is used.
> >     - Thus, the literature provides empirical evidence that our model justifies. Specifically, under the SSL Model, representations follow a mixture distribution (see Figure 2 in our paper) and are prevented from collapsing together, whereas discriminative SSL loss functions are minimised when each cluster collapses, losing information (as conjectured in [1]). A projection head mitigates this by taking representations from an "earlier" encoder layer to which the loss is not directly applied, which retains more variance and so more closely resembles the mixture prior.
> >     - Section 4.3.2 includes the above references.
> > - *“In SimVAE ... how is the number of clusters defined? Is a uniform prior assumed for p(y)”*:
> >     - In SimVAE, there is one cluster associated with each sample in the original dataset (as in Instance Discrimination and Contrastive Learning), e.g. for the set of augmentations of each image.
> >     - Yes, we assume a uniform distribution over $y$, the variable relating to "content" that is common across a set of semantically related data samples. Note that where images are augmented the same number of times, all classes/instances are equally likely to be sampled, justifying a uniform assumption.
> >     - Section 5 has been updated to make these points more clear.
> >
> >  [1] Chen, Ting et al. "A simple framework for contrastive learning of visual representations." International conference on machine learning. PMLR, 2020.
> >
> > [2] Wang, Yizhou et al. "Revisiting the Transferability of Supervised Pretraining: an MLP Perspective"
> >
> > [3] Gupta, Kartik et al. "Understanding and Improving the Role of Projection Head in Self-Supervised Learning"

---

> > > ### Author Response · Authors · 2024-07-23
> > > **Answer to Reviwer 43pU**
> > >
> > > - _“More discussion on fundamental insights to improve SSL”_
> > >     - Thank you for your comment. The proposed model and analysis provide new insight into the workings of popular self-supervised learning methods, e.g. demonstrating that style information can be lost, which may aid design of future SSL algorithms. Furthermore, generative representation learning seems a promising future direction offering:
> > >         - More informative representations: the SSL literature includes many tricks to avoid representation collapse, e.g. SimCLR, VicReg and MoCo, which are typically justified by their downstream performance. Our proposed SSL Model reframes these heuristics as approximations to an underlying principled "ideal" that inherently avoids collapse to learn representations that capture more information and potentially perform well on a broader range of downstream tasks, towards "task-agnostic" representations.
> > >         - Increased controllability over representations: the proposed SSL Model offers a clear interpretation of model parameters, contrary to traditional SSL. For example, the temperature in SimCLR remains a poorly understood parameter, whereas $\sigma_{p(z|y)}$ has a clear meaning in the proposed SSL Model. This opens up avenues for designing self-supervised models that learn representations with desired properties, such as disentanglement, through careful choice of the model's parameters.
> > >         - Uncertainty estimation: the SSL model provides an estimate of the uncertainty of representations through the posterior $q(z|x)$, contrary to discriminative SSL that assumes deterministic (encoder) mappings $f:\mathcal{X} \to \mathcal{Z}$. This has practical benefit, for example less confidence should be attributed to downstream predictions based on a representation with a high variance posterior $q(z|x)$.

---

### Review · Reviewer_WADv · 2024-07-14

**Summary Of Contributions:**

The authors propose a new generative model for self-supervised learning (SSL). The main idea is to assume the existence of two latent variables: one, $y$, that models the "content" of data points, and one, $z$ that governs the "style". For instance, different augmentations of the same image have the same content, but different styles.

The model can be trained using a VAE-style evidence lower bound (ELBO), that involves an encoder $q(z |x)$ that can be used as a learned representation.

The authors argues that several SSL objectives resemble their ELBO, generally with fewer terms. They perform a series of experiments on representation learning, and show that their reprentations compare savourably against SSL alternatives and other VAE models.

**Audience:**

Yes

**Broader Impact Concerns:**

I do not have particular concerns.

**Claims And Evidence:**

Yes

**Requested Changes:**

Clarifying the two points I mentioned in the "weaknesses" section.

**Strengths And Weaknesses:**

# Strenghts

- The paper is very well-narrated, and gives a excellent overview of modern SSL.

- The generative model is very sensible, and it is always compelling to have unifying perspectives on a topic.

- To the best of my understanding, the mathematical derivations seem sound.

- The experiments are convincing, albeit limited to smaller datasets than the ones used in most important SSL papers.

# Weaknesses

- One of the main claims is that most SSL losses are related to the ELBO of the proposed model. I feel like more details could be given to support this. It would help a lot to say explicitly, for each method, what is $z$, what is $y$, what is $J$, and what are the additional/missing terms. It is sometimes said clearly, and sometimes very unclearly (for deep clustering for instance).

- Modern SSL has been mostly influential for very large datasets, like Imagenet or Places (Zhou et al., Learning Deep Features for Scene Recognition using Places Database, NeurIPS 2014), but the authors consider smaller image datasets (MNIST, Fashion MNIST, CelebA, and Cifar10). This is not terrible, because training on such huge datasets is challenging, and requires expensive compute. However, the reasons for this choice should be more clearly explained, and this limitation should be more clearly acknowledged.



# Minor details

- This is a matter of personal preference, but I think that the notation that the authors use for integrals, namely $\int_x f(x)$, is not very popular, compared to more standard ones like $\int  f(x)dx$ or $\int  f(x) \mathrm{d} x$.

- typo in the abstract: Abstract: "self-supevised".

---

> ### Author Response · Authors · 2024-07-23
> **Answer to Reviewer WADv - Part 1**
>
> Thank you for your review and finding that our work is **well-written**, **relevant**, **compelling**, **sound** and has **convincing experimental results**; and also for your suggestions that have helped improve our work, which we address below and in the updated manuscript.
>
> - _“ ...more details to support the claim that most SSL losses are related to the ELBO of the proposed model, e.g. what is $z$, $y$, $J$, ..."_ :
>     - Thank you for your suggestion, we agree this could be more clear, particularly for deep/latent clustering. We have improved section 4.3 to make more clear that:
>         - $z$ is a representation of the data $x$, defined (in all cases considered) by an encoder $f: \mathcal{X} \rightarrow \mathcal{Z}$, e.g. the pre-softmax layers of a Resnet. Representations are used in downstream tasks, as described in section 2.
>         - $y$ is a high-level latent variable (referred to as content), taking a common value across semantically related data samples (e.g. image augmentations).
>             - In Instance discrimination (ID) and Contrastive learning (CL), $y=i$ is given by the index of a sample $x_i$.
>             - In Latent/Deep clustering,  for data point $x$, $y=c \in [1,C]$ is the cluster index assigned by the clustering method applied in latent space $\mathcal{Z}$.
>         - $J$ is the number of semantically related data samples (i.e. those having the same label $y$) considered simultaneously in the loss function.
>             - Instance discrimination and Latent/Deep clustering use a softmax layer to classify each sample individually according to its assigned label $y$ (as above), hence $J=1$.
>             - Contrastive learning considers *pairs* of semantically related data, hence $J=2$.
>     - *“additional/missing loss terms”*: the proposed ELBO encompasses three terms: entropy, the prior and a reconstruction term (see section 4.3).
>         - Only the entropy term is dropped from discriminative SSL objectives, due to their use of a deterministic encoder.
>         - The prior (marked purple) is included in Eqs 4 (for Instance Distrimination and Deep/Latent clustering) and 5 (for Constrastive learning).
>         - The reconstruction term is emulated in discriminative SSL (marked green/teal) following Eq 4 (for Instance Distrimination and Deep/Latent clustering) and Eq 5 (for Constrastive learning).

---

> > ### Author Response · Authors · 2024-07-23
> > **Answer to Reviewer WADv - Part 2**
> >
> > - _“larger datasets missing, explanation and clarification in text”_ :
> >     -  Thank you, we agree that it is important for representation learning methods to scale to large datasets, and that our choice of datasets should be well explained and limitations of our work made clear.
> >     - Firstly, the primary aim of our work is to propose a theroretical basis for families of self-supervised learning methods to give insight into how these methods work. Our analysis identifies that certain information ("style") may be lost by self-supervised methods, which our experiments set out to demonstrate. Our analysis also suggets that learning representations generatively by maximising the ELBO (rather than a discriminative proxy to it) may overcome this information loss, which we demonstrate on the same data sets.
> >     - However, learning representations for complex data distributions under a generative regime remains a challenge  relative to discriminative approaches. In particular, a generative model must capture sufficient information to fully regenerate an image, not only enough to discriminate samples; and must model complex distributions $p(x|z)$ in the (typically) high-dimensional data space. While the increased difficulty of generative modelling is well-known, recent work [1] quantifies this, showing that features used to succesfully predict labels may capture only a little of the variance in the data, which a generative model must capture in full. This demonstrates that discriminative models have an "easier job" relative to generative approaches and justifies why the latter may be less competitive on more complex datasets, e.g. ImageNet, given the same model architecture. Promisingly, [1] suggests ways in which these issues might be alleviated in the future and generative modelling has recently seen significant advances, e.g. diffusion models, suggesting that generative representation learning may greatly improve with future research.
> >     - Thus, our choice of data sets is chosen (i) to demonstrate the key findings of our analysis, in particular the information loss under discriminative methods and that a generative approach can learn useful representations, while (ii) being capable of being modelled by VAE-based models (the datasets we use, MNIST, FashionMNIST and CelebA, are reference benchmarks for VAE-based methods [e.g. 2, 3]).
> >     - In summary, we hope that by providing a theoretical rationale for self-supervised representation learning, which reveals pitfalls in currrent disciminative methods and suggests an alternative principled generative approach, we may encourage future research in the generative direction that is able to overcome the othogonal challenges a generative approach brings.
> >     - We have updated Section 8 to convey this more clearly.
> >
> > - Minor comments: thank you, we adjusted the manuscript accordingly.
> >
> > [1] Balestriero, Randall, and Yann LeCun. "How Learning by Reconstruction Produces Uninformative Features For Perception." Forty-first International Conference on Machine Learning.
> >
> > [2] Manduchi, Laura, et al. "Tree variational autoencoders." *Advances in Neural Information Processing Systems* 36 (2023): 54952-54986.
> >
> > [3] Sinha, Samarth, and Adji Bousso Dieng. "Consistency regularization for variational auto-encoders." *Advances in Neural Information Processing Systems* 34 (2021): 12943-12954.

---

### Author Response · Authors · 2024-07-23
**General response to rebuttal**

We thank all reviewers for their feedback and useful suggestions on our work. We address each comment in our rebuttal below and have revised the manuscript accordingly. All modifications are in red font in the updated manuscript uploaded to Openreview.

---

### Public Comment · ~Laurence_Aitchison1 · 2024-12-08
**Relevant prior work connecting VI with SSL**

Great work, and congrats on the acceptance!

Worth noting prior work in TMLR on an alternative approach to connecting SSL with VI.  Particularly showing that InfoNCE _is_ VI under a particular, but very odd, probabilistic generative model:

https://openreview.net/forum?id=chbRsWwjax

https://arxiv.org/abs/2107.02495

---

### Decision · Action_Editor_ALQb · 2024-08-27

**Recommendation:** Accept as is

**Comment:**

This paper provides an interesting theoretical insight for SSL. Their theoretical results look solid and well-motivated.

As many reviewers pointed out, one of the weaknesses could be the small-scale experiments. However, as the rebuttal clarified, the goal of this work is to propose a theoretical basis for families of SSL methods to give insight into how they work. It would be great if this paper could include more large-scale experiments, as with many previous SSL works, but I don't think it is necessary to convince the readers.

I think the current form of the paper sufficiently satisfies the TMLR evaluation criteria and recommend "Accept as is".

**Audience:**

Self-supervised learning (SSL) has a large number of audience. This work presents a theoretical framework for understanding SSL. I think this approach can be valuable to many researchers in this field.

**Claims And Evidence:**

This paper proposes a generative latent variable model for self-supervised learning (SSL) and provides a unifying theoretical framework for several families of SSL methods. More specifically, this paper shows that many SSL losses are related to the ELBO minimization of the proposed framework. Also, the analysis justifies the use of a projection head.

Based on this theoretical observation, this paper proposes generative self-supervised learning, named SimVAE. SimVAE is evaluated on several small-scale datasets, such as MNIST, FashionMNIST, CIFAR10 and CelebA.